# E-VAds: An E-commerce Short Videos Understanding Benchmark for MLLMs

Xianjie Liu [* † 1]  Yiman Hu [* ‡ 1]  Liang Wu [1]  Ping Hu [2]  Yixiong Zou [3]  Jian Xu [1]  Bo Zheng [1]

## Abstract

E-commerce short videos represent a high-revenue segment of the online video industry characterized by a goal-driven format and dense multimodal signals. Current models often struggle with these videos because existing benchmarks focus primarily on general-purpose tasks and neglect the reasoning of commercial intent. In this work, we first propose a **multi-modal information density assessment framework** to quantify the complexity of this domain. Our evaluation reveals that e-commerce content exhibits substantially higher density across visual, audio, and textual modalities compared to mainstream datasets, establishing a more challenging frontier for video understanding. To address this gap, we introduce **E-commerce Video Ads Benchmark**, which is the first benchmark specifically designed for e-commerce short video understanding. We curated 3,961 high-quality videos from Taobao covering a wide range of product categories and used a multi-agent system to generate 19,785 open-ended Q&A pairs. These questions are organized into Perception and Cognition and Reasoning, which consist of five distinct tasks. Finally, we develop **E-VAds-R1**, an RL-based reasoning model featuring a multi-grained reward design called **MG-GRPO**. This strategy provides smooth guidance for early exploration while creating a non-linear incentive for expert-level precision. Experimental results demonstrate that E-VAds-R1 achieves a 109.2% performance gain in commercial intent reasoning with only a few hundred training samples. Data is available at https://github.com/TaobaoTmall-AlgorithmProducts/E-VAds_Benchmark.

---

[*]Equal contribution , [†]Work done during an internship at Alibaba., [‡]Project leader.  [1]Alimama Tech, Taobao & Tmail Group of Alibaba [2]Vin University [3]Huazhong University of Science and Technology.  Correspondence to: Yiman Hu <huyiman.hym@alibaba-inc.com>, Liang wu <wuliang.wu@taobao.com.>, Jian Xu <xiyu.xj@taobao.com.>, Bo Zheng <bozheng@alibaba-inc.com>.

*Proceedings of the 43rd International Conference on Machine Learning*, Seoul, South Korea. PMLR 306, 2026. Copyright 2026 by the author(s).

## 1. Introduction

E-commerce videos have grown rapidly and now represent a major, high-revenue segment of online video. On e-commerce platforms (e.g. TikTok Shop, Amazon, Shopee, Taobao) (Si, 2021), videos are primarily designed to drive immediate purchases rather than general engagement, which makes their style distinct from other video content. In practice, these short ads are brief, fast-paced, and heavily edited, with a clear conversion goal. They pack dense multi-modal signals, such as rapid visual changes, on-screen text, continuous speech, and product close-ups and present them at the same time. This **multi-modally dense, goal-driven** format introduces new challenges for current video understanding.

Driven by advancements in LLMs, contemporary models excel at general video understanding. However, most efforts remain concentrated on general-purpose (Yu et al., 2019; Mangalam et al., 2023; Fu et al., 2025; Li et al., 2024; Tapaswi et al., 2016; Xiao et al., 2021) or long brand advertising (Long et al., 2025; Zhang et al., 2025b) videos, leaving the complex and high-value domain of e-commerce short videos largely under-explored. While existing benchmarks prioritize tasks like action recognition, commonsense QA, grounding and spatial-temporal relationships, they neglect the reasoning of commercial intent, such as selling points, target audiences, conversion strategies and so on.

This domain raises three practical challenges. **(1) High multi-modal information density:** models must track rapid visual changes while grounding dense speech and text overlays within short time windows. **(2) Benchmark gap:** there is still no dedicated benchmark that systematically evaluates conversion-oriented e-commerce short videos at scale. **(3) Open-ended commercial reasoning:** commercial questions (e.g., persuasion logic and consumer insight) are inherently open-ended, highly intent-driven and subjective, making supervision and evaluation less straightforward and often leading to sparse reward signals for learning.

To quantify the above "high infomation density" challenge, we further propose a **multi-modal information density assessment framework** with three complementary metrics: Visual dynamic density ($V_{den}$) (Zhang et al., 2025b), which captures the rate of semantic change over time to reflect transition and editing frequencies; Audio density ($A_{den}$), measured by the number of ASR words per unit of time

to represent speech intensity; and Textual density ($O_{den}$), defined by the frequency of OCR occurrences per frame to reflect the presence of on-screen text. As shown in Table 1, **E-VAds exhibits substantially higher density across vision, audio, and text than mainstream datasets, establishing a more challenging frontier**.

To fill this gap, we introduce **E-commerce Video Ads Benchmark (E-VAds)**, a benchmark for evaluating model performance on e-commerce short video understanding. We collect 3,961 high-quality videos from Taobao, covering a wide range of product categories, and apply a **dynamic sampling strategy** to improve category balance and annotation efficiency. Each video is converted into **a structured multi-modal context** that includes time-aligned ASR and OCR, visual evidence, and metadata. We then generate higher-quality question answering pairs using a **multi-agent annotation system** to reduce subjectivity in intent-based reasoning tasks for commercial videos. Multi-role agents propose and evaluate candidate QAs, and all items are further verified through rigorous manual review. The resulting benchmark contains 19,785 open-ended question answering pairs across five tasks, spanning two dimensions, *Perception* and *Cognition and Reasoning* (Figure 1).

Finally, we propose **E-VAds-R1**, an RL-based reasoning model to handle the modality-dense videos and the complex open-ended commercial questions. We design evidence-grounded rewards that encourage multi-modal attribution, and introduce **MG-GRPO**, a multi-grained reward design that ensembles reward granularities to provide **smooth guidance** during early exploration while creating a **non-linear incentive** for expert-level precision. With only a few hundred training samples, E-VAds-R1 achieves a significant 109.2% relative improvement in commercial intent reasoning over strong general-purpose baselines.

Our main contributions are as follows:

- We introduce **E-VAds**, the first benchmark for e-commerce short video understanding, with an automated construction pipeline to complement existing video benchmarks.
- We propose a **multi-modal information density assessment framework** and show that E-VAds contains much denser visual, audio, and textual information than mainstream datasets, making it more challenging for MLLMs' understanding.
- We develop **E-VAds-R1**, an RL-based reasoning model with a multi-grained reward design, achieving a 109.2% performance gain in the e-commerce domain.

## 2. Related Works

### 2.1. Video Question Answering Benchmarks

VideoQA benchmarks evaluate spatiotemporal understanding in videos. Representative datasets include human-centered benchmarks such as NextQA (Xiao et al., 2021) and MovieQA (Tapaswi et al., 2016), as well as instructional benchmarks such as EgoSchema (Mangalam et al., 2023) and VideoMME (Fu et al., 2025). However, these general benchmarks rarely capture the persuasive logic and conversion-oriented mechanisms central to advertising. While AdsQA (Long et al., 2025) and VideoAds (Zhang et al., 2025b) study advertising videos, they mainly focus on longer, carefully produced brand ads for brand awareness building, and largely overlook e-commerce short videos that target immediate conversion and contain tightly synchronized, multimodally dense signals.

### 2.2. Video Large Language Models

MLLMs build on vision language alignment from CLIP (Radford et al., 2021), with models such as LLaVA (Liu et al., 2023) and Flamingo (Alayrac et al., 2022) connecting vision encoders to LLMs for instruction following. This line has extended to video via Video-LLaVA (Lin et al., 2024), vidoe-LLama (Zhang et al., 2025a) and VideoChat (Li et al., 2025) using temporal aggregation for multi-frame modeling. Closed-source models including the GPT series (OpenAI., 2025) and Gemini (Google., 2025) further improve general multimodal reasoning, while InternVL (Zhu et al., 2025; Wang et al., 2025a) and QwenVL (Bai et al., 2025b;a) enhance fine-grained perception and long-context reasoning (Feng et al., 2026a). Recently, some models have also shown competitive performance, such as Keye (Team et al., 2025; Team, 2025a) and Mimo (Xiaomi, 2025). Despite this progress, existing open-source models still struggle with the high-density multimodal signals common in e-commerce videos.

### 2.3. Reinforcement Learning for Reasoning

In text-only settings, supervised fine-tuning is often constrained by data diversity, whereas RLHF (Achiam et al., 2023) better aligns outputs with human preferences. Recent results from DeepSeek-R1 (Shao et al., 2024) and OpenAI o1 (Jaech et al., 2024) suggest that reinforcement signals can strengthen reasoning. In advertising, AdsQA (Long et al., 2025) proposes ReAd-R, using an LLM-as-a-judge (Gu et al., 2024) to guide reflection on social intent. However, it mainly targets brand ads and focuses on metaphor and emotional tone, and does not emphasize the multimodal evidence needed for commercial intent reasoning in dense e-commerce videos such as those in E-VAds.

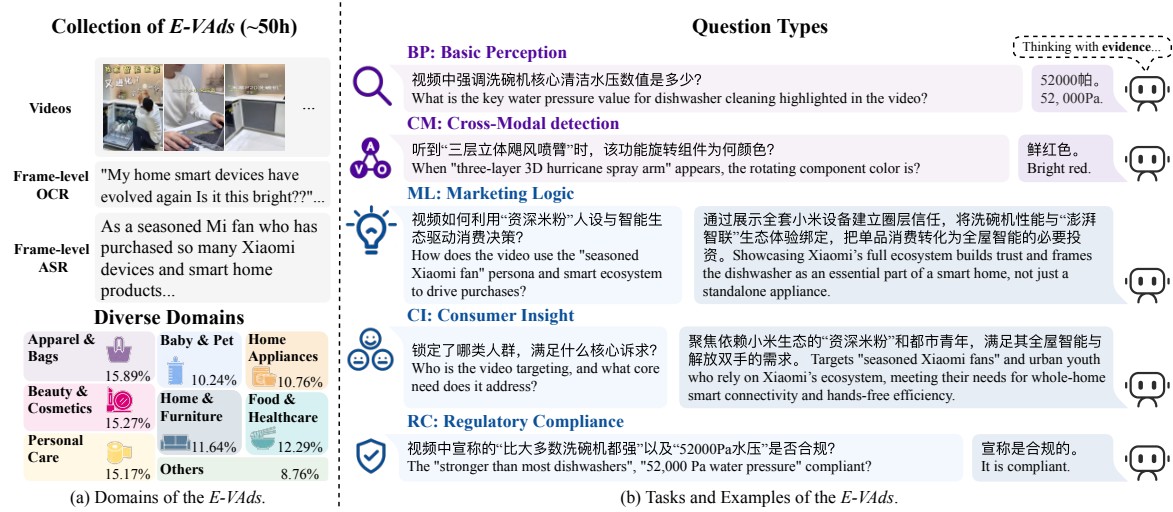

*Figure 1.* Overview of *E-VAds* benchmark.

*Table 1.* Comparison between E-VAds and other video benchmarks. **Anno** denotes annotation type (**M**: Manual, **A**: Automatic). **Task Types** include **MCQs** (Multiple-Choice Questions) and **Open**-ended QA. The best results are in **bold**, and the second best are underlined.

| Benchmark | QA nums | Anno | Task Types | $V_{\text{den}}$ | $A_{\text{den}}$ | $O_{\text{den}}$ |
|---|---|---|---|---|---|---|
| VideoMME-short (Fu et al., 2025) | 3,000 | M | MCQs | 31.65 | 1.16 | 4.32 |
| MVBench (Li et al., 2024) | 4,000 | M | MCQs | 20.69 | 2.46 | 3.45 |
| ActivityNetQA (Yu et al., 2019) | 8,000 | M | Open | 29.69 | 2.77 | 2.98 |
| EgoSchema (Mangalam et al., 2023) | 5,031 | A | MCQs | 25.76 | 0.00 | 4.24 |
| AdsQA (Long et al., 2025) | 7,895 | A+M | Open | 50.13 | 0.85 | 5.05 |
| VideoAds (Zhang et al., 2025b) | 1,100 | M | MCQs | 50.72 | 1.40 | 4.02 |
| **E-VAds (Ours)** | **19,785** | A+M | Open | **60.44** | **5.08** | **18.66** |
| *E-VAds-Test* (Ours) | 16,925 | A+M | Open | 60.48 | 5.09 | 18.81 |
| *E-VAds-RL* (Ours) | 980 | A+M | Open | 59.92 | 5.17 | 19.88 |
| *E-VAds-SFT* (Ours) | 1,880 | A+M | Open | 60.40 | 5.00 | 16.60 |

# 3. Multi-modal Information Density Assessment Framework

## 3.1. Definition of Multi-modal Information Density

To quantify information density and multi-modal complexity in e-commerce short videos, we define three modality-specific metrics that capture visual dynamics, spoken content, and on-screen text.

**Visual dynamic density** ($V_{\text{den}}$). Following (Zhang et al., 2025b), we use DINOv3-Base to extract frame features $f$. For a video with $T$ sampled frames, we compute the weighted average similarity of frame $i$ within a temporal neighborhood of size $d$:

$$\bar{S}_i = \frac{\sum_{j \in N_i, j \neq i} w(i,j) \cdot \cos(f_i, f_j)}{\sum_{j \in N_i, j \neq i} w(i,j)}, \quad (1)$$

where $\cos(\cdot)$ denotes cosine similarity and $w(i,j) = \exp\left(-\frac{|j-i|}{2d}\right)$ applies exponential temporal decay. We then define visual dynamic density as

$$V_{\text{den}} = \alpha \cdot \frac{1}{T} \sum_{i=1}^{T} \left(1 - \bar{S}_i\right), \quad (2)$$

where $\alpha$ is a scaling constant set to 100. A larger $V_{\text{den}}$ indicates more frequent visual changes and stronger editing dynamics.

**Audio density** ($A_{\text{den}}$) **and Textual density** ($O_{\text{den}}$). We define audio and textual density as the word counts of ASR and OCR content normalized by video duration:

$$A_{\text{den}} = \frac{|\mathcal{T}_{\text{asr}}|}{T}, \qquad O_{\text{den}} = \frac{|\mathcal{T}_{\text{ocr}}|}{T}, \quad (3)$$

where $\mathcal{T}_{\text{asr}}$ is the full-video ASR transcript, $\mathcal{T}_{\text{ocr}}$ is the concatenated OCR text from sampled frames, $|\cdot|$ denotes word count, and $T$ is the video duration (seconds). Larger $A_{\text{den}}$ and $O_{\text{den}}$ indicate denser speech and on-screen text, respectively.

Together, these metrics provide a unified view of how e-commerce ads present dense information across vision, audio, and text compared with general videos.

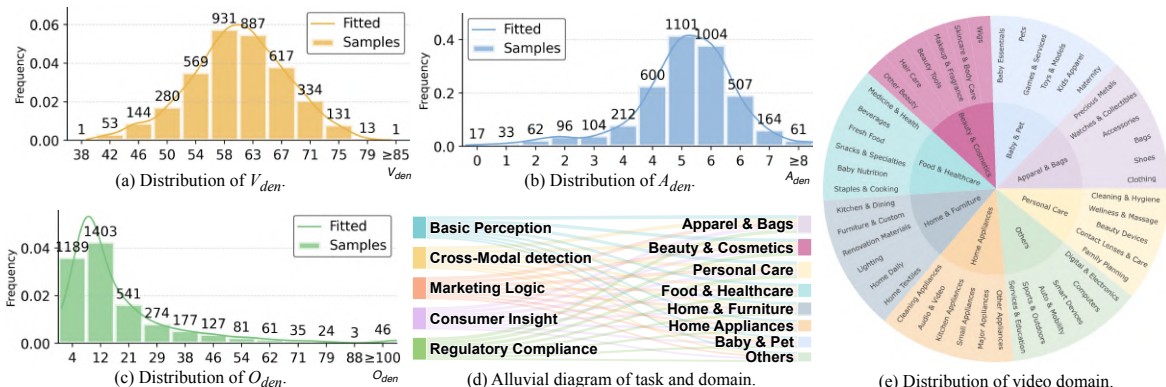

*Figure 2.* Statistics of E-VAds benchmark.

## 3.2. Metric Analysis and Comparison

Table 1 compares E-VAds with representative video understanding and advertising benchmarks in terms of scale, task format, and multi-modal information complexity. Across all three dimensions, E-VAds exhibits substantially higher density than both general video QA datasets and existing advertising benchmarks. Figure 5 and Appendix. A show the detailed distributions of the three density metrics in E-VAds.

These results confirm that e-commerce short videos are not a minor variant of general video QA, but **a qualitatively harder setting where models must operate under modality-dense and tightly synchronized signals.** In practice, models must *simultaneously* (i) track rapid visual changes, (ii) associate fast-evolving ASR/OCR cues with the correct visual evidence, and (iii) reason about commercial intent when signals are noisy, partially redundant, or even conflicting. Moreover, the metrics provide a principled way to quantify this difficulty and enable density-aware analyses, which were not provided in prior datasets.

## 4. The E-VAds Benchmark

To fill the gap in current video benchmarks, we introduce **E-VAds**, a benchmark designed to evaluate MLLMs' commercial understanding in conversion-oriented e-commerce short videos, which is an important and challenging domain.

In Section 3.1, we present fine-grained modality extraction, decomposition, and alignment to better handle high-density multi-modal signals. In Sections 3.2 and 3.3, to improve the reliability of the generated open-ended commercial-intent QA pairs, we adopt a multi-agent system for multi-round role-based generation and human expert review, and construct multi-modal evidence chain (Muennighoff et al., 2025; Guo et al., 2025) to support reasoning. The benchmark construction pipeline is shown in Figure 3 and the statistics are shown in Figure 5.

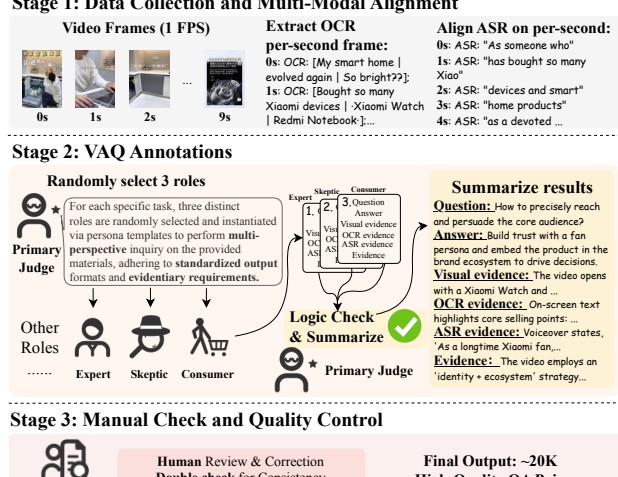

*Figure 3.* Dataset Construction Pipeline.

### 4.1. Data Collection and Multi-Modal Alignment

**Data collection and filtering.** To select high-quality data from billions of Taobao e-commerce advertising videos while covering all video categories as much as possible, we design an automated filtering pipeline that removes videos with weak commercial appeal, low-quality or overly short videos and samples with missing metadata, resulting in about 30,000 high-quality promotional videos.

We additionally propose a **dynamic sampling algorithm** to reduce annotation cost and alleviate category imbalance. Instead of uniform undersampling, we use a sigmoid-based function to set the sampling ratio $f(x)$ for each category:

$$f(x) = \frac{a}{1 + \exp\left(1 - \frac{b}{x}\right)}, \tag{4}$$

where $x$ is the original number of videos in a category, $a$ is the upper bound of the sampling ratio, and $b$ controls the curve's inflection point. *This function preserves minority*

*categories with higher sampling ratios while suppressing majority categories non-linearly*. After sampling, we obtain 3,961 videos with a more balanced category distribution (Fig. 2 (d)) and improved annotation efficiency.

**Dense multi-modal signal extraction, decomposition and alignment.** E-commerce short videos are produced to drive immediate conversions by coordinating persuasive cues across visual, audio, and textual modalities. To better leverage such carefully crafted videos for fine-grained analysis, we slice each raw video into **a compact, time-aligned event sequence** (Figure 3) that is easy for both annotators and models to inspect. We sample frames at 1 FPS, transcribe speech with Whisper-v3-large (Radford et al., 2022), and extract on-screen text with Qwen2.5-VL 32B (Bai et al., 2025b). We align all modalities on a 1-second timeline: OCR is assigned to its corresponding second, while each ASR segment is split into per-second non-overlapping chunks according to its temporal span (the last second absorbs remaining characters to avoid truncation). We then merge consecutive seconds with identical OCR and ASR content to reduce redundancy.

We formalize the structured context as

$$C = \{\langle f_t, \alpha_t, \gamma_t \rangle \mid t = 1, \ldots, T\} \otimes \mathcal{M}, \quad (5)$$

where $f_t$ is the visual keyframe feature at time $t$, $\alpha_t$ is the aligned speech text within $[t, t+1]$, $\gamma_t$ is the calibrated and de-duplicated OCR text, $\mathcal{M}$ is product metadata such as category and attributes, and $\otimes$ denotes temporal alignment and semantic concatenation across modalities. **This pipeline converts noisy multimodal streams into a structured evidence chain for downstream persuasive-content analysis.**

## 4.2. VQA Annotations

**Task definition.** E-commerce videos pack dense product details and promotional claims into a few seconds through coordinated visuals, narration, and on-screen text. As a result, models must first accurately recognize fine-grained concepts, and then reason about intent, audience, and compliance based on multi-modal evidence. Therefore, we design our tasks along two dimensions, perception and reasoning, to match the characteristics of e-commerce short videos. For perception, in addition to basic tasks that assess core recognition abilities, we introduce a cross-modal detection task to evaluate how well models handle the high-density multi-modal signals across modalities. For reasoning, we design three groups of tasks from the perspectives of advertisers, consumers, and platforms. In total, we define five task categories, summarized below, with detailed prompts provided in Appendix B.

Dimension 1: Perception.

- Basic Perception (BP): Identify product attributes and salient visual entities.
- Cross-Modal Detection (CM): Judge consistency and complementarity among ASR, OCR, and visual cues under noise.

Dimension 2: Cognition and Reasoning.

- Marketing Logic (ML): Unpack persuasive structure, including selling points and pain point to solution mapping.
- Consumer Insight (CI): Infer target audience from style, tone, and product characteristics.
- Regulatory Compliance (RC): Identify potential violations of advertising regulations.

**Multi-agent annotation system.** Due to the nature of e-commerce tasks, they are often open-ended and involve intent-driven subjective Q&As. To generate fairer and more objective Q&A pairs for such tasks, we design a multi-agent annotation system (Yuan et al., 2025) that arbitrates among responses from diverse roles and enforces supervision with **multi-modal evidence** refined from the time-aligned multi-modal sequence. During the annotation, each agent adopts a distinct commercial persona and proposes challenging, evidence-grounded questions based on the structured context $C$, while a primary judge moderates the process with support from multiple secondary roles. Detailed definitions are provided in Appendix C.

**Traceability constraint.** To ensure reproducibility and prevent impression-based answers, each QA must satisfy a strict traceability rule:

$$\texttt{Evidence(V, A, O)} \rightarrow \texttt{Reasoning} \rightarrow \texttt{Answer}.$$

Each answer must be supported by at least one evidence source: vision (V), ASR (A), or OCR (O).

**Cross-modal detection constraint.** For CM, we further enforce an *information-gap* design: one modality raises the question and another modality provides decisive evidence. This discourages single-modality shortcuts and explicitly tests cross-modal retrieval and alignment (Feng et al., 2026b).

**Question normalization.** We cap question length, remove leading phrasing and hints, and rewrite questions into concise and objective academic language so that performance depends on evidence-based inference.

## 4.3. Manual Check and Quality Control

After the automated annotation process, we implemented a review mechanism combining manual selection and a cycle-based elimination mechanism to ensure factual accuracy, clarity of expression, and consistency in difficulty. Appendix D shows the details of the review process, including the annotation interface, detailed reviewer checklists,

and complete manual verification specifications. After manual check and quality control, E-VAds ultimately contains 19,785 high-quality QA pairs from 3,961 videos.

# 5. The E-VAds-R1 Model

Motivated by the observed density-induced challenges and the open-ended nature of commercial reasoning, our subsequent E-VAds-R1 study further explores how to improve learning under sparse supervision by designing multi-grained rewards for reinforced fine-tuning.

Figure 4 shows the training framework of E-VAds-R1. Given a video and a question, the policy outputs <think> and <answer>, and a frozen LLM-as-a-judge provides a scalar reward to optimize the policy.

## 5.1. Data Splits and Output Format

The training set is split into E-VAds-Train-SFT (376 videos, 1,980 QA) for supervised instruction and format alignment, and E-VAds-Train-RL (196 videos, 980 QA) for reinforcement learning in complex commercial scenarios. The remaining 3,389 videos with 16,384 QA pairs form the E-VAds test split. All training samples follow a structured format: <think> and <answer>.

## 5.2. Training Pipeline

To better handle the complex commercial tasks in E-VAds, we use a two-stage pipeline from imitation to reinforcement learning. In the SFT stage, we convert E-VAds annotations into instruction-style samples that require explicit evidence grounding before answering, aligning the model with e-commerce semantics and enforcing the structured output. The resulting model learns basic e-commerce video understanding and cross-modal grounding. We then apply RL to improve attribution and reasoning consistency by rewarding outputs that are evidence-grounded, logically coherent, and explicitly link visual, ASR, and OCR cues to commercial intent. The prompting details are provided in Appendix F.

## 5.3. Reward Design

We use an LLM-as-a-judge (Gu et al., 2024; Xie et al., 2024; Chiang & Lee, 2023; Chen et al., 2024) to compare model predictions against expert-annotated ground truth (GT) and output a five-level score: $x \in \{0, 0.25, 0.5, 0.75, 1\}$. For each response, the judge verifies the answer against evidence from all three modalities: visual, ASR, and OCR, as well as the final evidence summary. We report three metrics:

- **Strict** ($S$): $S(x) = \mathbb{I}(x = 1)$.
- **Relaxed-3** ($R3$): $R3(x) = 1$ if $x = 1$; $R3(x) = 0.5$ if $x \in \{0.75, 0.5\}$; otherwise $R3(x) = 0$.
- **Relaxed-5** ($R5$): $R5(x) = x$.

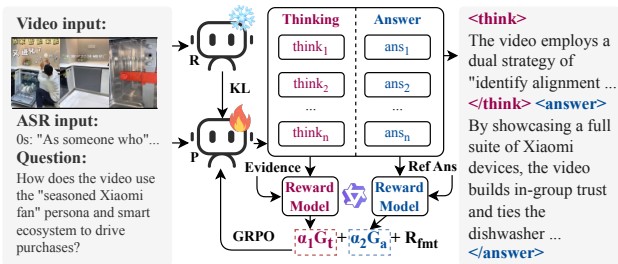

*Figure 4.* In the E-VAds-R1 framework, given a question, the policy model produces multiple responses including think and answer; these are scored by a reward model, and the resulting rewards guide policy updates through policy gradient optimization.

Here $\mathbb{I}(\cdot)$ is the indicator function. All metrics are averaged over samples. We provide the judge prompt, rubric, and examples in Appendix E.

During reinforcement learning, we score each generated trace along the following dimensions:

1. Reasoning trace: quality of the thinking ($x_t$).
2. Terminal answer: quality of the answer ($x_a$).
3. Format constraint ($R_{\text{fmt}}$): $R_{\text{fmt}} = -1$ if required tags are missing or malformed, otherwise 0.

**MG-GRPO.** To mitigate sparse rewards in open-ended commercial reasoning, we propose **Multi-Grained GRPO (MG-GRPO)**. It extends GRPO (Shao et al., 2024) by introducing a multi-grained reward mapping that calibrates responses across different levels of strictness:

$$G(x) = \frac{1}{3}\big[S(x) + R3(x) + R5(x)\big]. \tag{6}$$

By combining strict and relaxed scoring, $G(x)$ provides informative rewards for partially correct traces while still strongly favoring fully correct and well-grounded answers.

- Smooth Guidance for Exploration: By incorporating the relaxed metric $R5(x)$, $G(x)$ provides dense feedback for partially correct traces. For instance, a marginal improvement from $x = 0$ to $x = 0.25$ yields a non-zero reward ($G(x) \approx 0.083$), preventing the policy from being lost in a "zero-reward landscape" during early stages.
- Non-linear Incentive for Precision: The mapping creates an uneven reward landscape to penalize "near-misses." While the reward increment from $x = 0.5$ to $x = 0.75$ is relatively small ($\approx 0.084$), the leap from $x = 0.75$ ($G(x) \approx 0.417$) to a perfect trace ($x = 1, G(x) = 1.0$) is significantly larger. This non-linear jump, amplified by the strict metric $S(x)$, compels the model to pursue expert-level precision rather than settling for partially grounded reasoning.

The final reward $R$ is a weighted combination of the answer score ($x_a$), the reasoning trace score ($x_t$) and a format

*Table 2.* Benchmark results for different MLLMs. * means 400 randomly sampled questions.

| Model | Params | S | | | | | | R3 | | | | | | R5 | | | | | |
|---|---|---|---|---|---|---|---|---|---|---|---|---|---|---|---|---|---|---|---|
| | | BP | CM | ML | CI | RC | ALL | BP | CM | ML | CI | RC | ALL | BP | CM | ML | CI | RC | ALL |
| Expert* | - | .786 | .536 | .173 | .250 | .804 | .535 | .893 | .768 | .587 | .625 | .902 | .767 | .932 | .839 | .793 | .813 | .951 | .871 |
| **Closed-source Models** | | | | | | | | | | | | | | | | | | | |
| GPT 5.2 | - | .687 | .315 | .205 | .074 | .105 | .278 | .773 | .526 | .596 | .528 | .268 | .539 | .842 | .671 | .786 | .746 | .477 | .704 |
| Gemini3-Flash | - | .763 | .391 | .179 | .193 | .222 | .350 | .836 | .602 | .583 | .588 | .452 | .612 | .888 | .730 | .776 | .776 | .634 | .761 |
| **Open-source Models** | | | | | | | | | | | | | | | | | | | |
| Qwen3-Omni-Instruct | 30BA3B | .693 | .227 | .034 | .032 | .037 | .205 | .801 | .454 | .508 | .502 | .306 | .514 | .864 | .596 | .701 | .692 | .512 | .673 |
| Qwen3-Omni-Thinking | 30BA3B | .637 | .214 | .065 | .054 | .065 | .207 | .755 | .379 | .525 | .517 | .315 | .498 | .835 | .552 | .739 | .721 | .528 | .675 |
| VideoLlama3 | 7B | .306 | .082 | .000 | .000 | .000 | .078 | .448 | .208 | .259 | .286 | .049 | .250 | .578 | .382 | .397 | .404 | .232 | .399 |
| InternVL3 | 8B | .531 | .093 | .000 | .001 | .002 | .126 | .674 | .266 | .396 | .410 | .153 | .380 | .757 | .420 | .478 | .478 | .343 | .495 |
| InternVL3.5 | 8B | .522 | .084 | .004 | .005 | .003 | .124 | .664 | .268 | .483 | .480 | .146 | .408 | .759 | .423 | .629 | .621 | .347 | .556 |
| Keye-VL | 8B | .307 | .080 | .068 | .038 | .075 | .119 | .408 | .509 | .182 | .243 | .495 | .373 | .568 | .712 | .386 | .462 | .707 | .572 |
| Keye1.5-VL | 8B | .619 | .125 | .193 | .051 | .039 | .206 | .726 | .552 | .377 | .245 | .506 | .482 | .805 | .756 | .532 | .463 | .695 | .650 |
| Mimo-VL-SFT | 8B | .519 | .007 | .124 | .008 | .006 | .133 | .664 | .486 | .300 | .247 | .479 | .435 | .763 | .676 | .471 | .442 | .616 | .594 |
| Mimo-VL-RL | 8B | .600 | .146 | .009 | .007 | .010 | .155 | .734 | .349 | .491 | .490 | .258 | .464 | .814 | .514 | .698 | .648 | .463 | .628 |
| Qwen2.5-VL-Instruct | 32B | .550 | .150 | .001 | .003 | .003 | .142 | .704 | .354 | .485 | .490 | .289 | .464 | .799 | .523 | .695 | .680 | .478 | .635 |
| Qwen2.5-VL-Instruct | 7B | .446 | .108 | .000 | .000 | .001 | .111 | .608 | .268 | .374 | .390 | .206 | .369 | .714 | .437 | .458 | .468 | .368 | .489 |
| **E-VAds-R1(Qwen2.5-VL)** | **7B** | **.569** | **.127** | **.069** | **.021** | **.177** | **.193** | **.714** | **.315** | **.523** | **.496** | **.457** | **.501** | **.804** | **.497** | **.739** | **.711** | **.652** | **.680** |
| Qwen3-VL-Instruct | 30BA3B | .672 | .236 | .023 | .022 | .027 | .196 | .783 | .467 | .501 | .496 | .253 | .500 | .852 | .609 | .689 | .680 | .472 | .660 |
| Qwen3-VL-Instruct | 8B | .600 | .133 | .006 | .010 | .017 | .153 | .745 | .365 | .490 | .490 | .329 | .484 | .828 | .512 | .623 | .644 | .511 | .623 |
| Qwen3-VL-Thinking | 30BA3B | .566 | .144 | .011 | .007 | .012 | .148 | .711 | .342 | .497 | .490 | .262 | .460 | .803 | .517 | .717 | .661 | .473 | .634 |
| Qwen3-VL-Thinking | 8B | .605 | .172 | .051 | .051 | .051 | .186 | .723 | .349 | .519 | .514 | .307 | .482 | .810 | .523 | .738 | .717 | .514 | .660 |
| **E-VAds-R1(Qwen3-VL)** | **8B** | **.628** | **.252** | **.313** | **.126** | **.279** | **.320** | **.741** | **.427** | **.651** | **..555** | **.515** | **.577** | **.824** | **.592** | **.816** | **.761** | **.689** | **.736** |

constraint penalty:

$$R = \alpha_1\, G(x_a) + \alpha_2\, G(x_t) + R_{\text{fmt}}, \qquad (7)$$

where $\alpha_1 = 0.8$ and $\alpha_2 = 0.2$. Following GRPO (Shao et al., 2024), we sample $n$ traces $\{o_1, \ldots, o_n\}$ per prompt and compute the group-normalized advantage:

$$A_i = \frac{R_i - \text{mean}(\{R_1, \ldots, R_n\})}{\text{std}(\{R_1, \ldots, R_n\}) + \epsilon}. \qquad (8)$$

This multi-grained reward structure enhances group-relative discriminability: by providing a denser reward spectrum, it enables $A_i$ to capture subtle quality differences among traces within the same group, even when none are perfectly correct. This stabilizes the optimization and jointly emphasizes better reasoning paths and terminal answers especially for e-commerce video tasks.

## 6. Experiment

### 6.1. Experimental settings

To ensure reproducibility and fairness, we choose Qwen2.5-VL 7B Instruct and Qwen3-VL 8B Instruct as base models. Training uses 16 H20 GPUs with Llama-Factory (SFT) (Zheng et al., 2024) and ROLL (RL) (Wang et al., 2025b) Framework. SFT uses a batch size of 16 and RL uses a batch size of 12, with both performing a gradient update once per step. We set the learning rate to $1e{-}6$ for both SFT (10 epochs) and RL (2 epochs). We evaluate mainstream MLLMs under a unified protocol on our dataset. We use Qwen3-Coder-Plus (Team, 2025b) as the judge.

To comprehensively evaluate E-VAds, we will compare it with a range of SOTA multi-modal models, grouped into three categories: the current leading closed-source models, GPT-5.2 (OpenAI., 2025) and Gemini3-Flash (Google., 2025); general instruction-tuned models, including Qwen3-Omni (Xu et al., 2025), InternVL3 (Zhu et al., 2025), Keye-VL (Team et al., 2025), Mimo-VL (Xiaomi, 2025), VideoLlama3 (Zhang et al., 2025a), and the Qwen3-VL (Bai et al., 2025a) / Qwen2.5-VL (Bai et al., 2025b) series; and strong reasoning models, namely the thinking versions of Qwen3-Omni, InternVL3.5 (Wang et al., 2025a), Keye1.5-VL (Team, 2025a), Mimo-VL, and Qwen3-VL, which are designed to enhance logical reasoning capability. For GPT-5.2, we use 48 frames, while other MLLMs use 2 FPS.

Furthermore, to establish a human performance baseline, we randomly sampled 400 questions from the test set for manual evaluation. Two independent annotators were required to answer these questions by watching the videos only, without access to external ASR/OCR transcripts. For each question, annotators were tasked with providing both a terminal answer and the corresponding reasoning process within a strict 5-minute time limit. The final human perfor-

mance, denoted as "Expert*" in our results, is reported as the average score of these two annotators evaluated under the same LLM-as-a-judge protocol.

## 6.2. Results and Obervations

We comprehensively evaluate model performance on E-VAds using the strictness metrics defined in Section 5.3 and Qwen3-Coder-Plus as the judge, with results summarized in Table 2. We draw the following observations. And we give a case on Appendix. G.

**(a) E-VAds-R1 delivers the strongest improvement among open models by explicitly training reasoning.** E-VAds-R1 (8B) substantially improves over its base model, Qwen3-VL-8B, raising the **ALL** score from 0.153 to 0.320 under S, which is a **109.2%** relative gain. The largest gain is on RC, where E-VAds-R1 reaches 0.279 (S), about $16\times$ higher than the base model (0.017), and higher than GPT 5.2 (0.105) and Gemini3-Flash (0.222). This suggests that our RL training, which supervises the reasoning process, markedly improves commercial judgment.

**(b) Human experts remain a strong upper bound, highlighting the difficulty of commercial videos' reasoning.** Table 2 shows a substantial gap between SOTA MLLMs and human experts, especially on higher-order reasoning. Human experts achieve 0.535 (S) and 0.871 (R-5) on the **ALL** metric, while the best closed-source model, Gemini3-Flash, reaches 0.350 (S). The gap is largest on Marketing Logic (ML) and Consumer Insight (CI), where even the strongest models rarely exceed 0.200 under S, indicating that current models still lack the domain expertise that is needed to infer persuasion strategies and audience psychology in e-commerce ads.

**(c) Closed-source models lead overall but still fall short of human-level commercial understanding.** Closed-source models, including Gemini3-Flash and GPT 5.2, outperform most open-source baselines on E-VAds. Gemini3-Flash achieves the best **ALL** score at 0.350 (S) and 0.761 (R-5), and ranks first on CM with 0.391, suggesting stronger alignment between ASR narration and fast-changing visuals. GPT 5.2 is competitive but weaker than Gemini3-Flash, and both remain below human performance on commercial.

**(d) Standard instruction-tuned open-source models show a clear reasoning bottleneck that scaling alone does not fix.** Models such as VideoLlama3 (7B) and InternVL3 (8B) are adequate on Basic Perception (0.306 and 0.531) but collapse on ML and CI, where several score 0.000 under S. Scaling to 32B, as in Qwen2.5-VL, yields limited benefit and reaches only 0.001 on ML, suggesting that parameter scaling alone cannot solve the multi-step reasoning required in e-commerce. In contrast, thinking variants improve performance: Qwen3-VL-8B-Thinking increases the **ALL** S

*Table 3.* Ablation study of E-VAds-R1 based on Qwen2.5-VL-7B. S,R3,R5 means the different scoring method defined in Sec. 5.3

| EXP | SFT | Think | Answer | S | R3 | R5 |
|---|---|---|---|---|---|---|
| Baseline | – | – | – | .111 | .369 | .489 |
| a1 | A | – | – | .136 | .420 | .566 |
| a2 | T→A | – | – | .131 | .418 | .568 |
| b1 | – | – | S | .136 | .393 | .511 |
| b2 | – | – | R3 | .141 | .458 | .559 |
| b3 | – | – | R5 | .152 | .479 | .614 |
| c1 | T→A | – | R5 | .163 | .479 | .646 |
| c2 | T→A | – | G | .180 | .496 | .668 |
| d1 | T→A | $R5^{0.5}$ | $R5^{0.5}$ | .169 | .486 | .658 |
| d2 | T→A | $G^{0.5}$ | $G^{0.5}$ | .192 | .497 | .666 |
| d3 | T→A | $G^{0.2}$ | $G^{0.8}$ | **.193** | **.501** | **.680** |

score from 0.153 to 0.186, indicating the importance of structured reasoning.

## 6.3. Impact of different training strategies

We conduct comprehensive ablation studies to evaluate the impact of each training component, as shown in Table 3. The Baseline is Qwen2.5-VL 7B.

**(a) SFT Strategies.** We compare two primary SFT configurations: (a1) mapping questions directly to answers (A), and (a2) a thinking-based flow (T→A). Results show only marginal performance gaps between these variants. This suggests that the SFT stage primarily serves to align output formats and inject basic domain knowledge, whereas complex e-commerce reasoning capabilities are predominantly developed during the RL stage.

**(b, c) Reward Design.** During the RL stage, we evaluate reward functions with different strictness levels, including $S$, $R3$, and $R5$, as well as their ensemble variant $G$. Under single-granularity reward training, a more lenient reward consistently performs better than a stricter one (b3 >b2 >b1), suggesting that overly strict rewards produce sparse supervision in dense multi-modal settings and hinder exploration. Using the multi-granularity reward $G$ further improves performance (c2 >c1), indicating that our design can better handle the complexity of e-commerce video tasks.

**(d) Thinking and Weighting.** We further encourage the model to generate explicit reasoning process and reward it with multi-modal evidence. The results provide three insights. (i)The reasoning process improves commercial understanding (d1 >c1). (ii) Our multi-grained reward yields further gains under the reasoning paradigm, as ; and it provides complementary supervision at different strictness levels, which reduces reward sparsity while still enforcing evidence grounding, matching the dense and noisy signals in E-VAds videos. (iii) Assigning a larger weight to the answer score improves performance, as stronger answer-level

*Table 4.* Performance variation of models under adding ASR text.

| Model | Qwen3-Omini-Thinking | | | Keye1.5-VL | | |
|---|---|---|---|---|---|---|
| Metric | S | R3 | R5 | S | R3 | R5 |
| ASR | .092 | .346 | .549 | .092 | .338 | .536 |
| Video | .196 | .479 | .659 | .179 | .458 | .623 |
| Video+ASR | .207 | .498 | .675 | .206 | .482 | .650 |
| Δ ↑ | .060 | .040 | .024 | .152 | .052 | .044 |
| Model | Qwen3-Vl-8B-Thinking | | | E-VAds-R1 | | |
| Metric | S | R3 | R5 | S | R3 | R5 |
| ASR | .081 | .328 | .528 | .162 | .404 | .597 |
| Video | .160 | .458 | .634 | .257 | .532 | .701 |
| Video+ASR | .186 | .482 | .660 | .320 | .577 | .736 |
| Δ ↑ | .166 | .054 | .042 | .245 | .085 | .050 |

optimization prevents the model from producing plausible but weakly supported answers. Overall, these findings support our training and reward design choices and confirm that E-VAds is a challenging frontier in video understanding.

### 6.4. Impact of ASR and Video inputs

To assess the necessity of ASR transcripts and visual inputs for tasks in E-VAds, we conduct a comprehensive ablation study across four models, evaluating *ASR*, *Video*, and *Video+ASR* configurations, as detailed in Table 4. Our results reveal that the visual modality remains indispensable: model performance exhibits a clear progressive improvement when transitioning from *ASR* to *Video*, and further to *Video+ASR*. This key finding underscores the foundational role of visual information in the task. Furthermore, compared to *Video* inputs, incorporating ASR transcripts consistently yields substantial performance gains across all evaluated MLLMs, with the Omni model demonstrating the smallest improvement. Collectively, these results highlight that E-VAds effectively necessitates joint audio-visual reasoning, reinforcing its value as a benchmark for evaluating multi-modal evidence-based commercial understanding.

## 7. Conclusion

In this study, we present E-VAds, a comprehensive benchmark designed to evaluate multimodal large language models on conversion-oriented e-commerce short videos. These videos are characterized by rapidly synchronized visual, auditory, and textual cues. We constructed this benchmark through a rigorous pipeline that combines multi-agent automated generation with expert human verification. The final dataset contains 3,961 carefully curated videos that span diverse product categories. Accompanying these videos are 19,785 open-ended question-answer pairs. These evaluation items are systematically organized into five distinct task categories. The categories include Basic Perception, Cross-Modal Detection, Marketing Logic, Consumer Insight, and Regulatory Compliance. We quantified the in-

formational complexity of this domain through a dedicated multi-modal density assessment framework. The quantitative results demonstrate that E-VAds exhibits substantially higher visual dynamic density, audio speech intensity, and textual overlay frequency when compared to mainstream video understanding datasets. This empirical evidence establishes e-commerce video question answering as a uniquely challenging research frontier. The domain requires models to process tightly synchronized signals and execute multi-step commercial reasoning simultaneously. To address the supervision sparsity inherent in these open-ended commercial reasoning tasks, we developed E-VAds-R1. This reinforcement learning framework incorporates a multi-grained reward mechanism that strictly grounds model outputs in multi-modal evidence. The proposed MG-GRPO strategy delivers smooth optimization signals during early training phases. It also establishes a non-linear incentive structure that drives the model toward expert-level precision. Experimental evaluations confirm that E-VAds-R1 achieves remarkable performance improvements while utilizing only a few hundred training samples.

## Impact Statement

**Benchmark and research impact.** E-VAds provides a challenging, standardized benchmark for conversion-oriented e-commerce short videos under modality-dense signals, facilitating evaluation of MLLMs on fine-grained perception, cross-modal grounding, and commercial-intent reasoning.

**Practical impact.** The dataset and evaluation can support applications such as e-commerce video understanding, ad content analysis, retrieval and summarization, and assistance tools that help users better access key product information.

**Limitations and inherent video bias.** Our construction pipeline employs comprehensive bias mitigation strategies that encompass a multi-agent adversarial QA generation framework, strict cross-modal evidence traceability constraints, and rigorous human expert review. These measures effectively reduce biases that emerge during the annotation process. We acknowledge that inherent biases within the source e-commerce videos remain difficult to eliminate completely. Such characteristics encompass commercial promotion orientation, cultural and regional preferences, distinct creator presentation styles, and platform specific product selection patterns. These characteristics originate from the real-world nature of the commercial video content. Complete debiasing at the raw video level therefore remains an open technical challenge. We explicitly document this limitation to inform future research efforts that aim to develop more balanced and representative commercial benchmarks.

**Acknowledgement.** This work was supported by alibaba Group through Alibaba Research Intern Program.

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

# A. Distribution of datasets

We visualize the multi-modal information density distributions of E-VAds against VideoMME-S (Fu et al., 2025), ActivityNetQA (Yu et al., 2019), EgoSchema (Mangalam et al., 2023), MVBench (Li et al., 2024), VideoAds (Zhang et al., 2025b), and AdsQA (Long et al., 2025). E-VAds exhibits consistently higher multimodal information density as shown in Fig. 5 and Tab. 5. For $V_{den}$, its distribution is clearly shifted toward larger values than those of general-domain videos such as ActivityNetQA, which reflects faster editing and more frequent shot changes in e-commerce short videos. For $A_{den}$, E-VAds concentrates on the high-density region in terms of ASR word frequency, whereas the compared benchmarks are flatter or biased toward low-frequency regions, indicating that e-commerce videos contain substantially denser spoken content. For $O_{den}$, E-VAds shows the most pronounced advantage in OCR density because e-commerce videos frequently overlay stylized captions, key selling points, and price tags, which results in markedly more text per frame than non-advertising datasets.

*Table 5.* Comparison between our proposed E-VAds and other existing video benchmarks.

| Benchmark | Nums | Avg dur (S) | Total dur (H) | Avg size (MB) |
| --- | --- | --- | --- | --- |
| VideoMME-short | 300 | 80.9 | 6.64 | 9.87 |
| MVBench | 1,629 | 23.94 | 10.83 | 6.19 |
| ActivityNetQA | 574 | 107.8 | 17.06 | 23.77 |
| EgoSchema | 5,031 | 179.76 | 251.55 | 20.08 |
| AdsQA | 1,704 | 51.89 | 24.56 | 17.6 |
| VideoAds | 200 | 79.67 | 4.26 | 7.41 |
| E-VAds | 3,961 | 45.48 | 50.04 | 9.99 |

# B. Task category prompts

This section presents the core prompt logic that we use for automated annotation across five tasks as shown in Fig. 6, 7 and 8. For Basic Perception (BP), the prompt enforces objectivity by requiring the model to extract only physical attributes such as color, material, and numeric values, while explicitly forbidding subjective judgments. For Cross-modal Matching (CM), the prompt compels the model to ground references from ASR and OCR, such as deictic mentions like "this one", to a specific visual entity. For Marketing Logic (ML), the prompt follows a funnel-oriented analysis that guides the model to identify early hooks within the first few seconds, map USPs to user benefits, and recognize the design of calls to action. For Consumer Insight (CI), the prompt encourages backward inference by using cues from the scene context, presenter style, and background music to derive a concrete target-audience profile. For Regulatory Compliance (RC), the prompt implements a red-line and whitelist scheme that distinguishes permissible promotional rhetoric from illegal absolute claims, for example separating "miracle product" from terms such as "national-level" or "No.1".

# C. Details about multi-agent annotation system

We use a multi-agent collaboration framework to improve the quality and diversity of the generated QA pairs (Fig. 9 and 10). For secondary roles. We instantiate role-specific agents, where each agent asks questions that match its persona. We impose strict constraints such that each question is limited to 15 words and must not contain descriptive terms, which encourages abstract and challenging queries. Next, a primary judge agent aggregates raw evidence from all modalities, including visual content, OCR, and ASR. It consolidates multi-perspective observations into a single focused question and produces an explicit evidence chain that links multimodal cues to the final answer. In practice, to reduce annotation cost, we use strong Closed-source models. Secondary-role agents are instantiated with Gemini 3 Flash, while the primary judge is instantiated with Gemini 3 Pro.

**Multi-agent QA generation.** We generate QA pairs with a multi-agent collaborative annotation system. Each agent adopts a distinct commercial persona to simulate real business viewpoints and to convert the time-aligned evidence chain into challenging, evidence-grounded questions. Annotations are organized as a virtual round table led by a **primary judge** and supported by multiple secondary roles.

For perception-oriented tasks, annotators extract cues from five complementary perspectives with difficulty levels from *L1*

to *L3*:

- **Physical attributes**: color, material, shape, size, quantity, and motion.
- **Symbolic information**: brand, model, numbers, price, keywords, and contact information.
- **Relational evidence**: text–object grounding and vision–narration alignment.
- **Environmental context**: scene, location, weather, and landmarks.
- **Actionable behaviors**: operations such as unboxing, applying, pressing, stretching, step-by-step demonstrations, and user feedback (posture and facial expressions).

For reasoning tasks, we use personas with increasing difficulty from *L1* to *L5* (Higher levels indicate greater reasoning complexity):

1. **Consumer** (*L1–L3*): focuses on observable experience and perceived authenticity.
2. **Pragmatist** (*L2–L3*): emphasizes functionality, value, usage steps, price anchoring, and pain point matching.
3. **Skeptic** (*L2–L4*): checks for inconsistencies or missing information across vision, ASR, and OCR.
4. **Expert** (*L3–L5*): decomposes persuasive logic and marketing positioning.
5. **Creative Director** (*L4–L5*): analyzes audiovisual language and narrative structure.

## D. Manual Check and Quality Control

To ensure the rigor, precision, and commercial relevance of the E-VAds benchmark, we implement a multi-stage human verification pipeline as shown in Fig. 11. The process involves five professionally trained annotators and two senior researchers who serve as lead auditors to resolve disputes and perform final quality spot-checks.

Annotators evaluate each sample based on four fundamental pillars. First, *Accuracy* requires that answers remain strictly faithful to video facts and ASR/OCR references. Second, *Traceability*, which serves as the core principle, mandates that all responses be derived exclusively from provided multimodal evidence to prevent hallucinations based on external common sense. Third, *Discriminability* ensures that questions are sufficiently challenging such that they cannot be solved via shortcuts or linguistic biases without viewing the video. Finally, *Commercial Relevance* demands that answers reflect professional e-commerce insights, such as identifying a "price anchoring strategy" rather than merely describing a discount.

The verification workflow for the five core tasks (BP, CM, ML, CI, and RC) follows a recursive refinement logic. During the initial review, annotators assess the QA pairs and their corresponding evidence chains. While valid entries are immediately accepted, any item exhibiting logical inconsistencies or misaligned evidence triggers a multi-agent regeneration process. We enforce a cycle-based elimination mechanism that allows for a maximum of three regeneration attempts. If a task remains unsatisfactory after the third iteration, it undergoes manual correction by human experts to ensure the ultimate quality and integrity of the dataset.

This rigorous validation process ultimately produced approximately 19,785 high-quality QA pairs. The systematic approach effectively ensures the robustness, interpretability, and professional depth of each entry in E-VAds.

## E. LLM as Judge Prompt

Traditional lexical metrics such as BLEU (Papineni et al., 2002) and ROUGE (Lin, 2004) exhibit significant limitations in the context of e-commerce video understanding because they prioritize surface-level word overlap rather than semantic accuracy or business logic. These metrics often fail to distinguish between valid paraphrasing and factual hallucinations that arise from misinterpreting OCR or ASR metadata.

To address this, we introduce an LLM-as-a-judge (Gu et al., 2024) mechanism that simulates the perspective of a professional e-commerce analyst. This framework performs deep semantic verification by cross-referencing model outputs with video metadata, clues and ground-truth answers. By focusing on whether the provided evidence aligns with the actual video content, the judge effectively identifies logical disconnects that traditional metrics might overlook.

As illustrated in Fig. 12, the evaluator assigns a score from 0 to 1 based on five granular tiers:

- **1.0 (Perfect Match):** The response is accurate and professional with evidence that perfectly aligns with metadata.
- **0.75 (Accurate but Generic):** Core insights are correct but lack the depth of professional business analysis.
- **0.5 (artially Correct / Missing Info):** Only approximately half of the key points are captured or major background

facts are omitted.
- **0.25 (Logical Break / Misaligned Evidence):** The conclusion appears plausible but is supported by incorrect evidence which constitutes a factual hallucination.
- **0 (Completely Incorrect):** The response entirely deviates from the facts or fails to follow instructions.

This multi-tiered approach provides a more precise and objective assessment of MLLM performance in high-density information environments.

## F. Answer Prompts

For close-source models, we use this prompt as shown in Fig. 13 a. For reasoning models, we use this prompt as shown in Fig. 13 b. For instruct models, we use this prompt as shown in Fig. 13 c.

## G. Case Study

As show in Fig. 14, while general MLLMs often fail by being overly cautious or indecisive regarding regulatory nuances, E-VAds-R1 demonstrates expert-level judgment. It correctly distinguishes factual ingredient/origin claims from prohibited superlatives (e.g., "No.1" or "Best"), showing that our RL-based reasoning significantly bridges the gap between general perception and specialized commercial logic.

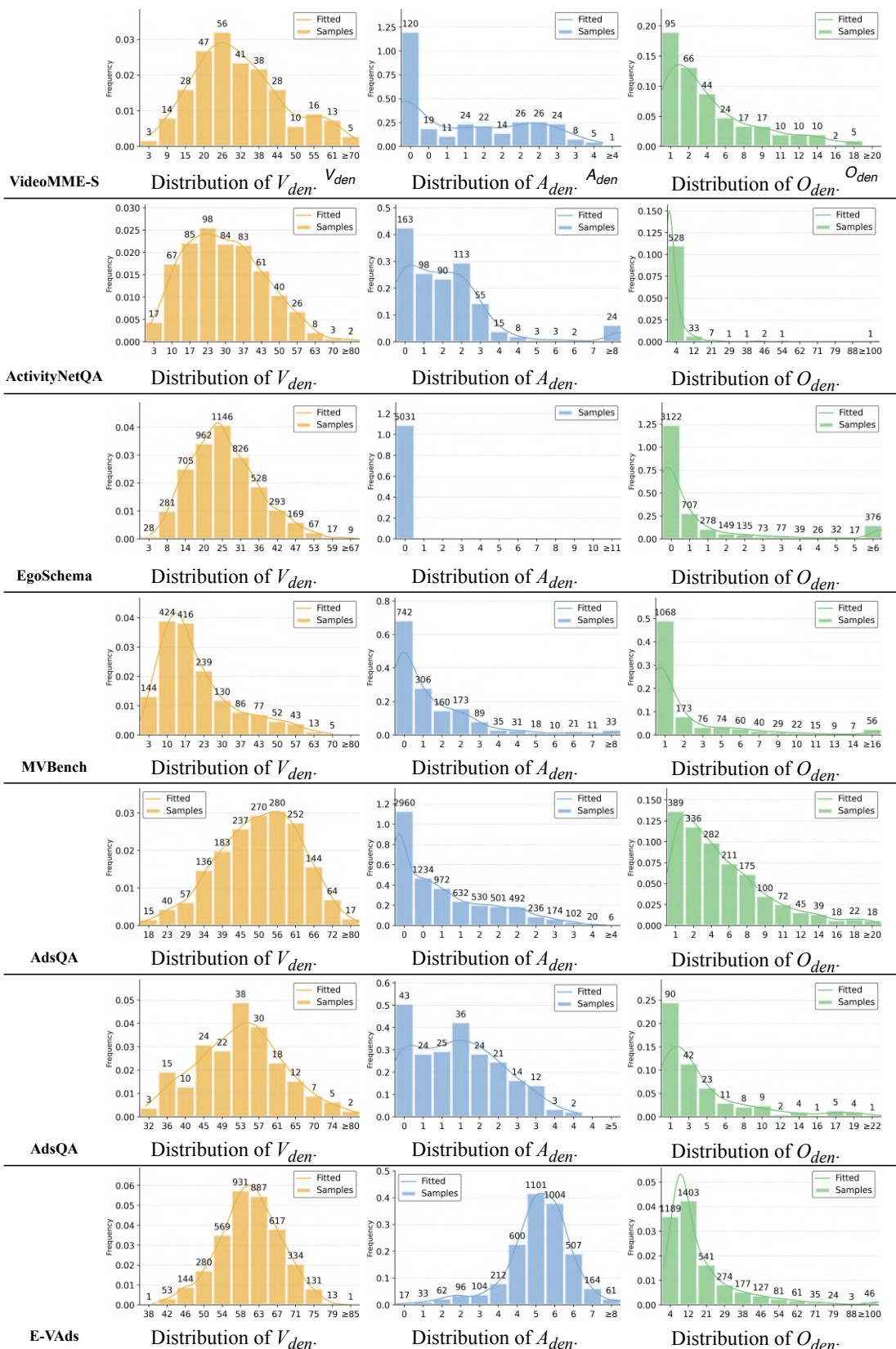

*Figure 5.* Detailed distributions of multi-modal information density metrics ($V_{den}$, $A_{den}$, and $O_{den}$) across datasets.

<Task category prompts>

## BP: Basic Perception

任务：多维基础事实提取（涵盖视觉、听觉、文本）。
核心逻辑：针对视频中的单一模态信息，进行客观、精确的物理特征描述或内容转录。
细分维度（题目需覆盖以下所有类别）：
1. 视觉-静态属性（Visual-Static）：颜色、形状、材质、纹理、数量、空间位置。
2. 视觉-动态变化（Visual-Dynamic）：操作动作（如按压、旋转）、物理状态改变（如起泡、变色）、展示方式。
3. 文本感知（OCR）：包装/字幕上的具体数值、品牌名、成分表、价格标签。
4. 听觉感知（ASR）：口播中提及的具体参数、特定的营销关键词、促销时间。
构造公式：[目标主体] + 的 + [特定感知维度] + 是什么/包含什么？
刚性约束：
1. 【客观性】答案必须是视频中可直接观测的物理事实，严禁包含"好用、高级、舒适、值得买"等主观评价或营销话术。
2. 【原子性】一个问题只针对一个具体的感知点，不要在一个问题里混合多个维度（如"颜色是什么且好不好看"）。
3. 【精确性】对于 OCR/ASR，必须精准匹配原文，不得改写。

Task: Multi-dimensional Ground Truth Extraction (covering visual, auditory, and textual modalities).
Core Logic: Provide objective and precise descriptions of physical attributes or transcriptions of content from a single modality within a video.
Sub-dimensions (questions must cover all categories below):
Visual-Static Attributes: color, shape, material, texture, quantity, spatial position.Visual-Dynamic Changes: operational actions (e.g., pressing, rotating), physical state changes (e.g., bubbling, discoloration), presentation methods.Text Perception (OCR): specific numerical values, brand names, ingredient lists, price tags appearing on packaging or subtitles.Auditory Perception (ASR): specific parameters mentioned in voiceovers, particular marketing keywords, promotional timeframes.
Question Construction Template:
"What is/What does [Target Subject]'s [Specific Perceptual Dimension]?"
(English equivalent: "What is/What does [Target Subject]'s [Specific Perceptual Dimension] ?")
Hard Constraints:
Objectivity: Answers must reflect only directly observable physical facts present in the video. Subjective evaluations or marketing language (e.g., "effective," "premium," "comfortable," "worth buying") are strictly prohibited.Atomicity: Each question must address only one specific perceptual point. Do not combine multiple dimensions in a single question (e.g., "What is the color and is it attractive?").Precision: For OCR/ASR content, responses must exactly match the original text or speech without any paraphrasing or modification.

## CM: Cross-Modal detection

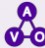

任务：针对电商短视频生成强依赖型多模态问答（VQA）测试集。题目必须构建一个逻辑闭环，使得模型若不理解 ASR/OCR 内容，就无法在复杂的画面中锁定正确的目标对象或时间点。
核心逻辑："文本索引 -> 视觉锁定 -> 视觉特征提取"。视频画面中必须包含多个相似物体、连续的多个动作或复杂的场景。文本（ASR/OCR）充当"过滤器"或"时间戳"，告诉模型去看哪一个。
构造公式：[ASR/OCR 中提及的特定条件/时间点/指代词] + 所对应/发生时刻的 + [画面中的特定对象] + 的 + [视觉物理属性/动作细节]？
刚性约束（防泄漏机制）：
禁止单品单场景：如果画面自始至终只有一个产品，禁止询问该产品的通用属性（因为这样不需要文本就能答）。
必须存在视觉干扰：问题指向的对象，必须在画面中有类似的干扰项（例如：画面有两双鞋，ASR 说"左边这双透气款"，问题必须针对"透气款"提问，迫使模型结合听觉判断左右）。
时间强相关：利用 ASR 的时间节点。例如"当主播提到'防水'的瞬间，水珠呈现什么状态？"（防止模型描述其他时间段的状态）。
回答规范：仅输出视觉观察到的事实，不包含推理性描述。

Task: Generate a strongly dependent multimodal question-answering (VQA) test set for e-commerce short videos. Each question must construct a logical closed loop such that the model cannot correctly identify the target object or time point within a complex visual scene without understanding the ASR/OCR content.
Core Logic: "Textual Indexing → Visual Localization → Visual Feature Extraction." The video frame must contain multiple similar objects, a sequence of multiple actions, or a complex scene. The textual information (from ASR/OCR) acts as a "filter" or "timestamp," instructing the model on which specific object or moment to attend to.
Question Construction Template:
[Specific condition / time reference / demonstrative pronoun mentioned in ASR/OCR] + 's corresponding / occurring-at-that-moment + [specific object in the visual scene] + 's + [visual physical attribute / action detail]?
Hard Constraints (Anti-Leakage Mechanisms):
• No single-product, single-scene videos: If the video contains only one product throughout, questions about generic attributes of that product are prohibited (since such answers would not require textual information).
• Visual distractors must exist: The object referred to by the question must have visually similar distractors in the scene (e.g., if two pairs of shoes appear in the frame and the ASR says "the breathable one on the left," the question must target "the breathable one," forcing the model to use auditory cues to resolve left/right ambiguity).
• Strong temporal dependency: Leverage ASR timestamps. For example: "At the exact moment the host mentions 'waterproof,' what is the state of the water droplets?" (This prevents the model from describing states from other time segments).
Answering Guidelines: Responses must state only visually observed facts, with no inferential or interpretive descriptions.

*Figure 6.* Task definitions and prompt instructions for Basic Perception (BP) and Cross-modal Matching (CM).

**\<Task category prompts\>**

## ML: Marketing Logic 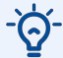

任务：剖析视频的转化逻辑与叙事结构。
核心逻辑：识别视频如何通过"吸引-说服-促单"的漏斗结构引导用户完成消费决策。重点在于分析"手段"而非单纯描述内容。
细分维度：
1. 黄金前三秒（Hook）：分析开头使用了什么策略（痛点场景重现、夸张视觉特效、悬念提问、利益点直给）来阻止用户划走。
2. 价值传递（USP）：识别视频是如何从"产品特征（Feature）"转化为"用户利益（Benefit）"的。
3. 心理博弈：识别促销手段（限时限量、价格锚点）、信任背书（专家/KOL推荐）、场景代入（解决具体烦恼）。
4. 行动号召（CTA）：分析结尾如何引导转化（话术引导、手势指引链接、评论区互动）。
构造公式：视频在 `[时间段/环节]` 使用了什么 `[营销策略/视听手段]`，目的是为了解决用户的什么 `[心理顾虑/需求]`？
刚性约束：
1.【透视本质】不要只回答"开头展示了衣服"，要回答"开头通过对比穿搭展示了显瘦效果，制造视觉冲击"。
2.【拒绝表面】分析必须关联到消费心理，例如提及"限时优惠"时，需指出这是为了利用"损失厌恶"心理。

Task: Analyze the video's conversion logic and narrative structure.
Core Logic: Identify how the video guides users through a "Hook → Persuade → Convert" funnel to drive purchase decisions—focusing on *strategies*, not just content description.
Dimensions:
1. Hook (First 3 Seconds): What tactic is used to stop scrolling? (e.g., pain-point reenactment, exaggerated visuals, suspenseful question, direct benefit claim)
2. Value Proposition (USP): How product features are translated into user benefits.
3. Psychological Triggers: Use of scarcity (limited-time/quantity), price anchoring, social proof (expert/KOL endorsement), or relatable scenarios solving specific problems.
4. Call-to-Action (CTA): How the ending drives conversion (verbal cues, gesture toward link, comment prompts).
Question Template:
In the "[time segment/phase]," what [marketing/audiovisual tactic] does the video use to address the user's [psychological concern/need]?
Hard Constraints:
1. Go Beyond Surface: Don't say "shows clothes"; say "uses side-by-side styling to visually demonstrate slimming effect for instant impact."
2. Link to Consumer Psychology: Always connect tactics to underlying motives—e.g., "limited-time offer" exploits loss aversion.

## CI: Consumer Insight 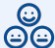

任务：受众画像侧写与场景匹配度分析。
核心逻辑：通过视频的视听语言（环境、模特、语气、BGM），逆向推其试图触达的目标人群及其生活方式。
细分维度：
1. 场景逆推：根据背景环境（宿舍/豪宅/办公室/户外露营）判断目标生活场景。
2. 人群侧写：根据模特形象、穿搭风格及语言风格，定义核心受众（Z世代学生、精致宝妈、商务男士等）。
3. 痛点对齐：评估视频提出的解决方案是否精准击中该人群的真实痛点（而非伪痛点）。
4. 情绪共振：判断视频营造的氛围（焦虑、治愈、亢奋、高级感）是否符合目标客群的心理预期。
构造公式：基于画面中的 `[视觉/听觉线索]`，该视频锁定的核心受众是 `[具体人群]`，试图满足他们的 `[特定情感/功能需求]`？
刚性约束：
1.【具体化】人群标签不能笼统（如"年轻人"），需细化（如"追求性价比的大学生"或"注重穿搭的职场新人"）。
2.【依据充分】推断必须基于视频内的客观线索，不能凭空臆想。

Task: Audience Profiling and Scene-Target Alignment Analysis
Core Logic: Reverse-engineer the intended target audience and their lifestyle by analyzing the video's audiovisual language (setting, talent, tone, BGM).
Dimensions:
Scene Inference: Deduce target life context from background environments (e.g., dorm, luxury home, office, camping).Audience Profiling: Define core demographic based on talent appearance, styling, and verbal tone (e.g., Gen Z students, style-conscious new professionals, busy working moms).Pain Point Alignment: Assess whether the solution addresses a genuine need of that group—not a fabricated one.Emotional Resonance: Evaluate if the mood (anxiety, calm, excitement, premium feel) matches the audience's psychological expectations.
Question Template:
Based on [visual/auditory cues] in the video, the core target audience is [specific demographic], aiming to fulfill their [specific emotional or functional need].
Hard Constraints:
Be Specific: Avoid vague labels like "young people"; use precise descriptors (e.g., "budget-conscious college students" or "image-aware entry-level professionals").Evidence-Based: All inferences must be grounded in observable elements from the video—no speculation.

*Figure 7.* Task definitions and prompt instructions for Marketing Logic (ML) and Consumer Insight (CI).

<Task category prompts>

## RC: Regulatory Compliance 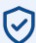

任务：电商合规性风控与尺度判别。
核心逻辑：在"营销修辞"与"实质违规"之间进行精准划界。区分"夸张的广告创意"与"欺诈性的虚假宣传"。
判别原则（白名单与红线）：
1.【体感豁免（白名单）】：涉及"保暖、亲肤、柔软、显瘦、透气"等主观感受词，若有画面佐证（如拉扯面料、上身对比），视为合规营销描述。
2.【修辞包容（白名单）】：允许使用"神器、天花板、绝绝子、闭眼入"等电商高频情绪词，不判定为违规极限词。
3.【数据宽容（白名单）】：非医疗/功能性强的普通商品（如服饰日用），不强制要求第三方检测报告，符合常识即可（如羽绒服说保暖）。
4.【绝对化红线（违规）】：严禁"国家级、最高级、第一、唯一"等广告法禁用词。
5.【虚假承诺红线（违规）】：严禁违背科学常识的断言（如"穿上立马瘦20斤"、"永久不坏"）。
构造公式：视频中提到的 `[特定文案/声明]` 属于 `[合规修辞/违规风险]`，理由是 `[判别依据]`？
刚性约束：
1.【结合语境】必须结合视频画面判断。如果主播说"弹力很大"，并用力拉扯衣服展示回弹，则判定为合规。
2.【严谨定性】指出违规时，必须引用具体的违规类别（如"虚假承诺"或"绝对化用语"）。

Task: E-commerce Compliance Risk Assessment and Boundary Judgment
Core Logic: Precisely distinguish between permissible marketing hyperbole and actual violations—differentiating creative exaggeration from fraudulent false claims.
Guidelines (Whitelist vs. Red Lines):
Sensory Claims Exempt (Whitelist): Subjective terms like "warm," "skin-friendly," "soft," "slimming," or "breathable" are compliant if visually supported (e.g., fabric stretching, before/after wearing).Rhetorical Expressions Allowed (Whitelist): Common e-commerce hype phrases like "game-changer," "best-in-class," "absolutely amazing," or "no-brainer buy" are not considered illegal superlatives.Data Leniency for Non-Functional Goods (Whitelist): Ordinary products (e.g., apparel, daily necessities) don't require third-party test reports; claims aligned with common sense (e.g., "down jacket keeps you warm") are acceptable.Absolute Language = Violation (Red Line): Banned terms like "national-level," "#1," "best," "only," or "topmost" violate advertising law.False Promises = Violation (Red Line): Claims defying scientific consensus are prohibited (e.g., "lose 20 lbs instantly just by wearing it" or "lasts forever").
Question Template:
The statement "[specific claim/script]" in the video is classified as [compliant rhetoric / high-risk violation] because [reasoning based on guidelines].
Hard Constraints:
Context Matters: Always evaluate claims alongside visual evidence. E.g., saying "super stretchy" while demonstrating strong, elastic recovery is compliant.Precise Violation Typing: If flagged as non-compliant, specify the exact violation category (e.g., "false promise" or "absolute language").

*Figure 8.* Task definitions and prompt instructions for Regulatory Compliance (RC).

## \<Multi-agent prompts\>

**Secondary roles**

系统：你当前身份是 {role_type}。
人物画像：{role['desc']}
目标难度：{role['level_target']}
提问风格：{role['style']}
你所属的分类重点如下：
{instruction}
视频元数据脚本（上下文）：
{self.context}
你的任务：
生成 1 组专业的 QA 对。
- 问题（Question）：必须是极简抽象提问（限15字内）。严禁包含手段、表现形式或视频细节。
- 视觉证据：场景/动作描述（visual_evidence）：结合视频场景/动作描述。
- OCR 证据：屏幕文本（ocr_evidence）：结合视频 OCR 文本。
- ASR 证据：旁白内容（asr_evidence）：结合视频 ASR 文本。
- 回答（Answer）：在此处进行简要回答。像真人一样直观地给出结论。想象你在评价一个作品，直接说明它的优劣或问题所在。禁止提及任何模态术语（如 OCR/ASR/视觉）。
- 证据（Evidence）：深度分析，需整合视觉、文本和音频信息。
规则：
- 极简抽象化：问题严禁超过 15 个字。禁止在问题中出现"通过..."、"利用..."、"基于..."、"视听配合"、"画面"、"台词"等描述手段的词汇。
- 去描述化：问题中不得出现视频里的具体事物或修饰性形容词。
- 核心导向：问题应直指广告的终极目的（如：受众定位、品牌心智、合规风险、转化逻辑），将所有"为什么"和"怎么做的"细节全部留在回答中。
- 禁止时间戳：问题和回答中不得提及具体秒数。
- 模态协同（环环相扣）：
- 问题的设计必须确保：如果不综合视觉、OCR 和 ASR 这三者，就无法在【回答】中给出完整的逻辑链条。
- 结构化证据：证据（evidence）字段结合三大支柱输出：
1. 视觉证据(visual_evidence)：对特定场景/动作的描述。
2. OCR 证据(ocr_evidence)：屏幕上显示的特定文本内容。
3. ASR 证据(asr_evidence)：旁白的直接引用或转述。
- 字数限制：像真人一样直观地给出结论。想象你在评价一个作品，直接说明它的优劣或问题所在。禁止提及任何模态术语（如 OCR/ASR/视觉）。回答应自然地将证据编织在一起（中文 30 字左右）。
安全规则：严禁涉及毒品、烟草、赌博或暴力。内容需对 18 岁以下青少年友好。
输出格式：JSON {{"question": "...","answer": "...""visual_evidence": "...","ocr_evidence": "...","asr_evidence": "...","evidence":"...",}}

System: Your current role is {role_type}.
Persona: {role['desc']}
Target Difficulty: {role['level_target']}
Questioning Style: {role['style']}
Your primary focus area:
{instruction}
Video Metadata Script (Context):
{self.context}
Your Task:
Generate 1 professional QA pair.
Question: Must be an ultra-concise, abstract question (≤15 words). Do not include methods, formats, or video-specific details.Visual Evidence: Describe relevant scenes/actions from the video.OCR Evidence: Quote specific on-screen text.ASR Evidence: Paraphrase or quote voiceover content.Answer: Provide a brief, intuitive judgment as if evaluating a creative work—state its strengths, flaws, or issues directly. Do not mention modalities (e.g., OCR/ASR/visual).Evidence: In-depth analysis integrating visual, textual, and audio information.
Rules:
Ultra-Concise & Abstract: Questions must be ≤15 words. Avoid phrases like "through...", "using...", "based on...", "audio-visual synergy", "visuals", or "dialogue".Non-Descriptive: No specific objects or adjectives from the video in the question.Core-Focused: Target the ad's ultimate intent (e.g., audience targeting, brand perception, compliance risk, conversion logic). Keep all "why" and "how" details in the answer.No Timestamps: Never reference specific times.Modality Synergy: The question must require combining visual, OCR, and ASR evidence to fully answer.Structured Evidence: The evidence field must synthesize:
Visual evidence: scene/action descriptionOCR evidence: on-screen textASR evidence: voiceover content
Answer Length: ~30 Chinese characters—natural, intuitive, and modality-agnostic.
Safety: No drugs, tobacco, gambling, or violence. Content must be suitable for under-18 audiences.
Output Format: JSON {{"question": "...","answer": "...""visual_evidence": "...","ocr_evidence": "...","asr_evidence": "...","evidence":"...",}}

Other Roles ······ **Expert**  **Skeptic**  **Consumer**

*Figure 9.* Prompt structure for secondary-role agents in the multi-agent system.

## <Multi-agent prompts>

### Primary role

**Primary Judge**

系统：你面前有{len(candidate_questions)}个不同视角针对主题"{task_category}"的初步观察。
你所属的分类重点如下：
{instruction}
【任务目标】
参考{len(candidate_questions)}个视角的分析，生成一个直观且有力的提问。
你必须全量整合{len(candidate_questions)}个视角的原始证据（视觉、OCR、ASR）并给出逻辑推导。
【背景元数据】
{self.context}
【已有视角的原始分析（包含证据）】
{detailed_inputs}
【生成准则】
- 提问风格（关键平衡）：提问要利落、见血见喉，旨在用该问题测试模型能力。
- 简要化：问题20个字左右。禁止出现描述手段词（如：通过、基于、视觉、听觉）。
- 证据全量整合：
- visual_evidence: 解答问题所需的视觉线索。
- ocr_evidence: 解答问题所需的OCR文字线索。
- asr_evidence: 解答问题所需的ASR文字线索。
- 深度逻辑链（evidence）：在此字段中，解释回答的逻辑推导和证据。
- 专业回答（answer）：直接给出正确回答。禁止提及模态术语。禁止回答任何问题未涉及的信息。字数控制在 50 字以内。
【输出格式】
必须返回 JSON 格式：
{{"question": "","answer": "...""visual_evidence": "...","ocr_evidence": "...","asr_evidence": "...","evidence": "...",}}

System: You are presented with {len(candidate_questions)} preliminary observations from distinct perspectives on the topic "{task_category}".
Your primary focus area:
{instruction}
[Task Objective]
Based on the analyses from all {len(candidate_questions)} perspectives, generate one intuitive and incisive question.
You must fully integrate the original evidence (visual, OCR, ASR) from all perspectives and provide a coherent logical inference.
[Background Metadata]
{self.context}
[Existing Perspectives with Raw Evidence]
{detailed_inputs}
[Generation Guidelines]
Question Style (Critical Balance): Sharp, penetrating, and designed to rigorously test model reasoning capability.Conciseness: The question should be around 20 Chinese characters. Do not use methodological terms (e.g., "through", "based on", "visual", "audio").Full Evidence Integration:
visual_evidence: Visual cues necessary to answer the question.ocr_evidence: On-screen text required for the answer.asr_evidence: Spoken content essential to the answer.
Logical Chain (evidence): Explain the reasoning process by synthesizing all evidence.Professional Answer (answer): State the correct conclusion directly. Do not mention modalities or introduce information beyond the question scope. Keep within 50 Chinese characters.
[Output Format]
Return strictly in JSON format:

*Figure 10.* Integrated prompt for the primary judge agent in the multi-agent system.

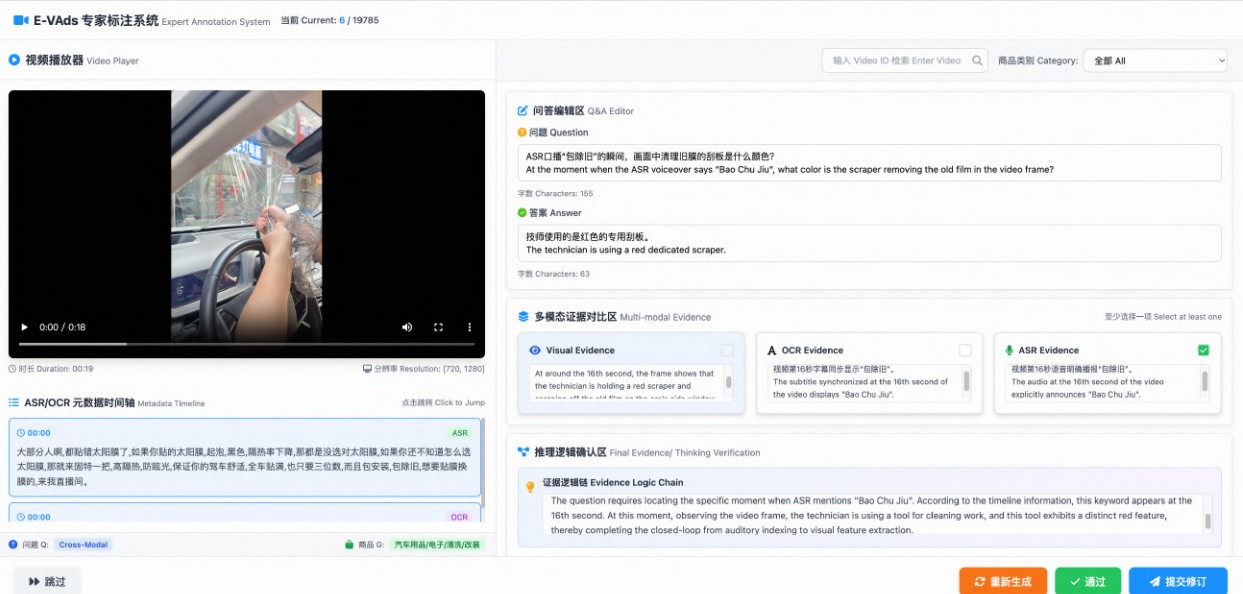

*Figure 11.* E-VAds Annotation System.

## <LLM As Judge prompt>

**LLM as Judge** 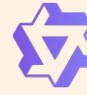

# Role
You are a professional E-commerce Content Analyst and Senior Evaluator. Your task is to objectively and rigorously assess the quality of an answer generated by a Large Language Model (LLM) based on multi-source inputs.

# Task
Compare the [Model Output] with the [Reference Answer], while considering the [Clues] and [Metadata (OCR/ASR)]. Assign a score from 0 to 1 based on the provided evaluation criteria.

# Evaluation Criteria
Please strictly follow these scoring tiers:

- 1.0 (Perfect Match):
- The answer is completely accurate and covers all key points of the reference answer.
- Evidence cited (e.g., specific text or lines) aligns perfectly with OCR/ASR records.
- Demonstrates deep business insight with professional terminology and logic.

- 0.75 (Accurate but Generic):
- The core answer is correct, evidence is accurate, and all key points are covered.
- Weakness: The response is too broad or lacks the deep logic/professional E-commerce perspective found in the reference (e.g., simply saying "it's attractive" without explaining the underlying conversion logic).

- 0.5 (Partially Correct / Missing Info):
- The direction of the answer and evidence is correct, but information is incomplete.
- Weakness: Only captures about 50% of the key points, or oversimplifies complex issues by missing important background facts from the metadata.

- 0.25 (Logical Break / Misaligned Evidence):
- Displays a clear "Right/Wrong Mismatch":
- Case A: The cited OCR/ASR evidence is real, but the conclusion derived is entirely wrong.
- Case B: The conclusion seems correct, but the cited evidence (timestamps, quotes, etc.) does not exist in the metadata (Hallucination).

- 0 (Completely Incorrect):
- The answer directly contradicts the OCR/ASR facts.
- Or, the model failed to answer the question or produced severe hallucinations.

# Input Data
1. [Question]: {question}
2. [Metadata (OCR/ASR)]: {metadata}
3. [Clues]: {clues}
4. [Reference Answer]: {reference_answer}
5. [Model Output]: {model_output}

# Output Template
Answer:

# Constraints
- DO NOT include any introductory text.
- DO NOT provide a range. You must choose ONE exact value from the scale.
- The Answer line must contain "Answer: " and the number.

*Figure 12.* Evaluation prompt and scoring rubric for LLM-as-a-Judge.

---

### (a) <The prompt template for Closed-source Models>

**Video**: {*video (or video frames for GPT5.2)*}
**Voicer**:{*asr*}
**Question**: {*question*}
You are permitted to gather any relevant clues, think step by step, to answer this question.
Output your thought process (no length limit) and the final answer (using approximately 50 words) in the following format
<think>your_thinking</think>
<answer>your_answer_within_50_words</answer>

---

### (b)<The prompt template for Reasoning Models>

**Video**: {*video*}
**Voicer**:{*asr*}
**Question**: {*question*}
You are permitted to gather any relevant clues, think step by step, to answer this question.
Output your thought process (no length limit) and the final answer (using approximately 50 words) in the following format
<think>your_thinking</think>
<answer>your_answer_within_50_words</answer>

---

### (c)<The prompt template for Instruct Models>

**Video**: {*video*}
**Voicer**:{*asr*}
**Question**: {*question*}
By watching the video, you are required to answer a question within 50 words:

*Figure 13.* Prompt Format for Inference.

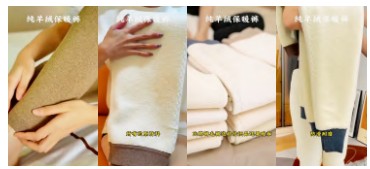

**RC Question:**
Do the phrases "all raw materials sourced from Inner Mongolia" and "pure cashmere" in the video constitute prohibited absolute claims?

**GT:**
No. They're factual origin/ingredient claims, not prohibited absolutes—fully compliant.

**Qwen3-VL 8B Instruct:**
Yes. "All from Inner Mongolia" and "pure cashmere" are absolute claims that may constitute exaggerated advertising—use with caution.
**LLM as Judge -> Score: 0**

**Qwen3-VL 8B Thinking:**
If the product genuinely uses 100% Inner Mongolia-sourced materials and is pure cashmere, it's compliant. However, claims like "all" or "pure" require solid evidence—without it, they risk being deemed false advertising. Verify ingredient authenticity.
**LLM as Judge -> Score: 0.5**

**E-VA-R1 (Qwen3-VL 8B):**
No. "Pure cashmere" and "all from Inner Mongolia" are factual claims—if substantiated, they're compliant, not prohibited superlatives. No "No.1" or "best"-type language used.
**LLM as Judge -> Score: 1**

**E-VA-R1 (Qwen3-VL 8B) Thinking Process:**

1. The video clearly showcases the pantyhose's "pure cashmere" texture and close-up shots of the cashmere lining (from 1s to 1.5s), accompanied by voiceover and on-screen subtitles, emphasizing the factual background that the raw materials originate from "herders' households in Inner Mongolia."

2. The phrase "pure cashmere," when factually accurate, constitutes a legitimate description of the product's material composition. Combined with sensory descriptors like "soft and delicate," it aims to convey key selling points—such as warmth and skin-friendly comfort—without employing prohibited absolute terms like "ultimate," "best," or "strongest."

3. In this context, "pure cashmere" is not an implausible or deceptive claim but rather a straightforward statement of the product's fundamental attribute. Similarly, the claim "entirely sourced from Inner Mongolia," as a traceable origin statement, qualifies as reasonable marketing language—provided it is supported by visual evidence (e.g., footage of herders harvesting cashmere or authentic on-location shots)—and does not constitute a prohibited superlative such as "the only one nationwide" or other "national-level highest-grade" assertions banned under advertising regulations.

4. Conclusion: The described statements represent reasonable and factual promotion of product attributes and do not constitute prohibited "false promises" or unlawful use of absolute advertising language.

**Qwen3-VL 8B Thinking Process:**
The question asks whether the statements "All raw materials come from Inner Mongolia" and "Pure cashmere" in the video constitute violations of absolute advertising terms under Chinese advertising regulations.

First, I need to analyze whether these phrases qualify as absolute claims that could be misleading or unlawful under China's Advertising Law.

China's Advertising Law requires all advertising content to be truthful, accurate, and non-misleading. The use of absolute terms—such as "all" or "pure"—can be problematic if they are unsubstantiated, exaggerated, or inconsistent with the actual product composition.

1. "All raw materials come from Inner Mongolia":
    This statement asserts that 100% of the raw materials originate exclusively from Inner Mongolia. If this claim is factually accurate—that is, no raw materials are sourced from any other region—then it complies with advertising regulations. However, if even a portion of the materials comes from elsewhere, the statement becomes false or misleading and would violate the law.

2. "Pure cashmere":
    The term "pure cashmere" typically implies that the product is composed of 100% cashmere fiber, with no blending of other materials (e.g., wool, synthetic fibers). Under Chinese standards (such as GB/T 16986-2018 for textile fiber content labeling), a product labeled as "pure cashmere" must contain at least 95% cashmere, with the remaining 5% allowed only for unavoidable impurities (not intentional blending). If the product meets this standard, the claim is acceptable. If it contains intentionally added non-cashmere fibers beyond allowable thresholds, the claim would be deceptive and thus a violation.

To determine whether these statements violate advertising rules, the following must be verified:
- Whether the raw materials are indeed sourced entirely from Inner Mongolia.
- Whether the product meets the legal definition of "pure cashmere."

Without specific evidence about the product's sourcing and composition, a definitive legal conclusion cannot be drawn. However, under Chinese regulatory practice, advertisers bear the burden of proof for such absolute claims. If the advertiser can substantiate both claims with credible documentation (e.g., supply chain records, third-party testing reports), then the statements are lawful. If not, they constitute violations.

Given that the video appears to be promotional content, it is reasonable to assume the advertiser intends the claims to be accurate. Nevertheless, regulatory authorities would require verification. In many enforcement cases, unverified absolute claims like "all" or "pure" have been deemed misleading and penalized.

Conclusion:
The statements "All raw materials come from Inner Mongolia" and "Pure cashmere" are not inherently violations—they are permissible if and only if fully substantiated by factual evidence. If the product genuinely meets both conditions, the claims comply with Chinese advertising law. If not, they constitute violations of prohibitions against false or misleading advertising. Absent contradictory evidence, and assuming standard compliance practices by the advertiser, these statements are likely acceptable. However, strict adherence to substantiation requirements remains essential.

*Figure 14.* Case Study.

