# OpenReview forum: "E-VAds: An E-commerce Short Videos Understanding Benchmark for MLLMs"
_ICML.cc/2026/Conference — ICML 2026 regular_

### Official Review · Reviewer_mnV7 · 2026-03-02

**Soundness:** 3
**Presentation:** 3
**Significance:** 2
**Originality:** 3
**Overall Recommendation:** 3
**Confidence:** 4

**Summary:**

This paper targets testing and improving MLLMs’ capabilities of understanding e-commerce short videos with dense multimodal information. The authors introduce E-VAds, a benchmark of 3,961 Taobao short ads with 19,785 open-ended QA pairs across five tasks spanning perception and higher-level commercial reasoning (e.g., marketing logic and regulatory compliance). They propose multimodal information density metrics to quantify domain difficulty and develop E-VAds-R1, an RL-based reasoning model with a multi-grained reward (MG-GRPO) design. Experiments show that domain-specific RL significantly improves performance on commercial reasoning tasks over strong baseline MLLMs.

**Compliance With Llm Reviewing Policy:**

Affirmed.

**Ethical Review Concerns:**

All videos are from Taobao.

**Ethical Review Flag:**

Flag this paper for an ethics review.

**Ethics Expertise Needed:**

["Discrimination / Bias / Fairness Concerns"]

**Key Questions For Authors:**

1. What’s the results of directly using all training data for SFT?

**Limitations:**

See weakness

**Strengths And Weaknesses:**

**Strength:**

1.	High practical relevance: the E-Vads benchmark has potential impact for industry applications (ad auditing, compliance checking, marketing analysis).

2.	Challenging multimodal reasoning frontier: By quantifying and demonstrating high modality density, the dataset pushes beyond current perception-heavy benchmarks.

3.	The paper is generally well-structured and easy to read.

**Weakness:**

1.	Limited domain scope: The benchmark is specialized to e-commerce ads, which may limit broad adoption compared to general VideoQA benchmarks.

2.	Domain generalization: All videos are sourced from Taobao. It is unclear whether the learned reasoning generalizes to other platforms (e.g., TikTok, Amazon). Cross-platform validation would strengthen claims.

3.	Potential annotation subjectivity: Despite multi-agent arbitration, tasks like Marketing Logic and Consumer Insight remain inherently subjective. Inter-annotator agreement statistics are not discussed.

4.	Incremental RL design: MG-GRPO is conceptually close to existing GRPO and reward smoothing approaches. The main novelty lies more in domain adaptation than in fundamental RL innovation.

---

> ### Author Rebuttal · Authors · 2026-03-31
>
> > 1. Limited Domain Scope Specialized To E-commerce Ads
>
> While E-VAds is specialized, we contend this focus is a core strength because e-commerce videos pose a **qualitatively harder problem** than general videos, as Section 3 quantifies their substantially higher multimodal information density that demands sophisticated reasoning capabilities which current datasets inadequately test. Second, this domain holds **significant practical value** since e-commerce short videos dominate as a high-revenue format on global platforms, making models that understand their persuasive logic critical for content analysis, recommendation, and summarization with direct economic impact. Finally, E-VAds **reveals critical weaknesses in advanced MLLMs** since our experiments show top-tier models struggle with commercial reasoning, proving strong general performance does not guarantee competence in specialized scenarios, providing a crucial tool to diagnose gaps and advance commercially-aware models. We will clarify these points in revision to highlight our work's broader impact.
>
> > 2. Domain Generalization
>
> We evaluated our model on VideoAds, an external commercial video understanding benchmark. Results show E-VAds-R1 generalizes well:
> |Model|Total Accuracy|Finding Accuracy|Reasoning Accuracy|Summary Accuracy|
> |-|-|-|-|-|
> |Qwen2.5-VL-7B|54.73%|62.59%|50.69%|48.72%|
> |ReAd-R|54.27%|60.24%|47.66%|53.85%|
> |E-VAds-R1 (Ours)|55.82%|63.29%|50.41%|51.92%|
>
> E-VAds-R1 achieves the highest total accuracy (55.82%) with strong Finding Accuracy (63.29%), validating that capabilities learned from E-VAds transfer effectively to other commercial video tasks. These improvements are notable given VideoAds has different data distribution and annotation style than E-VAds, demonstrating our training approach's robustness.
>
> > 3. Annotation Subjectivity
>
> E-VAds' human review uses a professionally qualified team rather than casual scoring:
>
> 1. **Personnel:** Five professional annotators conduct reviews, with two senior researchers as lead auditors for final arbitration
> 2. **Process:** A cycle-based elimination mechanism triggers up to three regenerations for samples with logical disputes, after which experts correct or discard unsatisfactory ones
> 3. **Traceability:** All subjective inferences anchor to objective visual or textual evidence
>
> This cross-validation mechanism ensures the dataset's objectivity and quality.
>
> > 4. Incremental RL Design
>
> We thank the reviewer. While MG-GRPO builds on work like GRPO, our contribution is a **comprehensive framework** comprising: (1) a novel high-density commercial video benchmark with five task categories, (2) rigorous multi-agent data construction with traceability constraints, (3) novel multi-modal information density metrics (Vden, Aden, Oden), and (4) the MG-GRPO training method designed for this domain.
>
> Focusing solely on MG-GRPO overlooks foundational challenges. Table 1 shows E-VAds exhibits the highest multi-modal information density among benchmarks (Vden: 60.44 vs. 29.69 for ActivityNetQA), creating unique optimization challenges that standard RL methods fail to solve:
> |Reward Design|S|R3|R5|Key Characteristic|
> |-|-|-|-|-|
> |R5|0.163|0.479|0.646|Smooth but encourages satisficing|
> |MG-GRPO (Ours)|0.180|0.496|0.668|Balances exploration & precision|
>
> MG-GRPO's multi-grained formulation addresses the fundamental dilemma in commercial reasoning regarding **how to balance initial exploration with expert-level precision**. The relaxed component (R5) provides dense signals during cold-start (G(0.25) ≈ 0.083), while the strict component (S) creates a non-linear jump for perfect answers (G(1.0) = 1.0 vs. G(0.75) ≈ 0.417). This structured solution prevents reward saturation in high-density, multi-faceted reasoning tasks rather than mere smoothing.
>
> The 109.2% relative improvement of E-VAds-R1 over baselines (Table 2) validates that this **integrated approach** drives substantial progress. We believe this multi-grained reward design is generalizable to other complex reasoning domains where output quality is nuanced.
>
> > 5. Results of Using All Training Data for SFT
>
> We have initiated the required experiments. Notably, due to substantial GPU resource demands for SFT training and limited rebuttal time, we could not complete full-data SFT training and evaluation within this period. Current training loss trends resemble those of single-data SFT with similar values, suggesting that simply increasing SFT data yields few performance gains. We commit to supplementing complete experimental results and detailed comparative analysis in the final manuscript.
>
> > 6. Ethical Review Concerns
>
> We have obtained proper authorization to use Taobao video data for research purposes, and the dataset will be released under licensing agreements that respect platform terms of service.

---

> > ### Author Rebuttal · Reviewer_mnV7 · 2026-04-04
> >
> > I appreciate the authors' careful rebuttal. Most of my concerns are addressed. However, I still think that E-commerce is a specialized domain that more data for SFT could be helpful and RL is not well motivated. Unfortunately, the expected experiment "using all training data for SFT"  cannot be obtained at the rebutall period.  I thus maintain my original score.

---

> > > ### Author Response · Authors · 2026-04-04
> > >
> > > We sincerely thank the reviewer for your constructive feedback.
> > >
> > > **Following the first round of rebuttal**, we conducted the additional experiments requested: SFT on the combined dataset (SFT + RL data). The complete results are presented below:
> > >
> > > | Training Strategy     | Data              | S         | R3        | R5        |
> > > | --------------------- | ----------------- | --------- | --------- | --------- |
> > > | SFT (table3 a1)       | sft data          | 0.136     | 0.420     | 0.566     |
> > > | **SFT (new)**         | **rl data**       | **0.129** | **0.414** | **0.562** |
> > > | **SFT (new)**         | **sft + rl data** | **0.138** | **0.432** | **0.585** |
> > > | Zero RL (table3 b3)   | rl data           | 0.152     | 0.479     | 0.614     |
> > > | E-VAds-R1 (table3 d3) | sft + rl data     | 0.193     | 0.501     | 0.680     |
> > >
> > > As shown, SFT on the full dataset yields **only marginal improvements** over SFT on SFT data alone (0.566 → 0.585). In contrast, even Zero RL on RL data significantly outperforms all pure SFT variants (0.614 > 0.585).
> > >
> > > This substantial gap underscores the necessity of RL for our specific task. **The E-VAds QA dataset involves complex reasoning rather than simple pattern matching.** While SFT focuses on likelihood maximization, **RL enables the model to optimize directly for logical consistency and final answer correctness through trial-and-error exploration.** This capability is critical for navigating the intricate reasoning paths required in e-commerce advertising scenarios, which simple data scaling via SFT cannot adequately address.
> > >
> > > We hope these additional results adequately address your concerns. We would be deeply grateful if you would consider raising your score in light of this new evidence.

---

### Official Review · Reviewer_fxMx · 2026-03-12

**Soundness:** 2
**Presentation:** 3
**Significance:** 4
**Originality:** 4
**Overall Recommendation:** 4
**Confidence:** 3

**Summary:**

The paper introduces E-VAds, a novel benchmark tailored for the e-commerce short video domain. The authors first establish a quantitative framework to demonstrate that e-commerce content possesses significantly higher multi-modal information density (visual, audio, and text) compared to general video datasets. The benchmark comprises thousands of videos with diverse Q&A pairs covering Perception and Cognition & Reasoning. Additionally, the authors propose E-VAds-R1, utilizing a Multi-Grained reinforcement learning approach (MG-GRPO) to enhance commercial intent reasoning.

**Compliance With Llm Reviewing Policy:**

Affirmed.

**Final Justification:**

I will still go with my earlier decision weak accept.

**Key Questions For Authors:**

1. Could the authors clarify the intuition or empirical evidence behind the selection of reward components ($S, R_2, R_5$) and their respective weights? Is there a specific reason why these particular metrics were prioritized over others?
2. Regarding the findings in Section 6.1 (d), how should we interpret the relatively high performance of Keye-VL models in ML/CI tasks? Does this suggest that certain pre-training data or architectural designs are more effective at bypassing the "reasoning bottleneck" mentioned in the text?

**Limitations:**

It would be highly beneficial if the authors could discuss the potential influence of the specific e-commerce ecosystem from which the E-VAds benchmark is sourced. Since marketing styles, regulatory standards, and consumer interaction patterns vary significantly across global markets, explicitly addressing how these regional factors might impact the generalizability of the findings to different cultural advertising contexts would strengthen the paper.

**Strengths And Weaknesses:**

Strengths:
1. The paper addresses "Commercial Intent Reasoning," a critical real-world application with high commercial value. It successfully bridges the gap between general video understanding and domain-specific industrial needs.
2. The multi-modal density assessment framework is an insightful contribution for quantifying video complexity. The "traceability constraint" used in dataset construction is a commendable approach to ensure data quality and grounding.
3. The overall narrative is well-structured and the motivation is clearly articulated. The visualization of information density effectively highlights the unique challenges of the e-commerce domain.

Weaknesses
1. The evaluation of the proposed MG-GRPO is currently incomplete. The manuscript would be significantly strengthened by including direct performance comparisons with standard RL baselines (e.g., GRPO, DPO or PPO). Additionally, the design choices for reward components ($S, R_2, R_5$) and their associated hyperparameters would benefit from empirical justification or ablation studies to demonstrate their specific impact.
2. The conclusions in the "Thinking and Weighting" section regarding the role of answer-level weights appear to be based on a limited number of comparative experiments. More extensive evidence across diverse categories would be necessary to statistically support the claim that such weighting prevents weakly supported answers.
3. In Section 6.1 (d), the authors discuss a reasoning bottleneck that scaling alone might not resolve. However, the relatively strong performance of Keye-VL/Keye1.5-VL in Table 3 appears to present an interesting case that might not fully align with this hypothesis at first glance. It would be very helpful if the authors could provide more nuanced analysis or discussion to help reconcile these observations and clarify the underlying factors.
4. The manuscript contains several typographical errors and occasional inconsistencies in internal cross-referencing (e.g., mismatched Figure/Table numbers and Section citations), which should be rectified for the final version.

---

> ### Author Rebuttal · Authors · 2026-03-31
>
> > 1. Incomplete MG-GRPO Evaluation
>
> Our baseline is standard GRPO, which outperforms PPO and DPO on numerous tasks [1,2]. While GRPO removes the Critic network, it faces severe reward sparsity in open-ended commercial QA. MG-GRPO addresses this by combining continuous and discrete rewards, which provides better exploration stability with high memory efficiency.
>
> Ablation results comparing reward designs:
> |Method|S|R3|R5|
> |-|-|-|-|
> |Baseline(Qwen2.5-VL-7B)|0.111|0.369|0.489|
> |GRPO|0.163|0.479|0.646|
> |MG-GRPO|0.193|0.501|0.680|
>
> Results show MG-GRPO with multi-grained rewards significantly outperforms single-granularity baselines, which validates our reward design.
>
> > 2. Limited Evidence for Answer-Level Weights
>
> Weight selection was based on empirical intuition from open-ended reasoning [3,4]. In E-VAds, the final answer must be grounded in multi-modal evidence (Visual, Audio, OCR). We hypothesized that higher answer weight would penalize plausible answers lacking evidence support, which prevents hallucinated reasoning that fails to reflect video content. This forces the model to prioritize evidence-grounded accuracy over superficial fluency.
>
> To validate this hypothesis, we conducted a simple reweighting experiment comparing d2 (equal weights: $G^{0.5}/G^{0.5}$) against d3 (answer-weighted: $G^{0.2}/G^{0.8}$). The results in Table 3 show consistent improvements when increasing answer weight from 0.5 to 0.8 (S: 0.192→0.193, R3: 0.497→0.501, R5: 0.666→0.680), confirming our intuition that stronger answer-level optimization encourages tighter evidence grounding.
>
> Due to the short rebuttal period, we could not conduct large-scale hyperparameter ablations since RL training requires significant GPU resources. We commit to include complete ablation results in the final manuscript.
>
> > 3. Keye-VL/Keye1.5-VL Performance Analysis
>
> Keye1.5-VL outperforms other open-source models due to two key factors:
>
> **1. Domain-Aligned Training Data.** Keye models are trained by Kuaishou (Kwai), which is a TikTok-like and TaoBao-like short video platform. Their training data contains massive short-form video content, which creates natural domain bias to our data.
>
> **2. Advanced RL Training (SAPO).** The difference between Keye-VL and Keye1.5-VL lies in RL approaches. While Keye-VL uses standard GRPO, Keye1.5-VL employs SAPO, which assigns different weights to dialogue stages for better understanding. This step-level weighting shares the same insight as MG-GRPO, which both recognize that different output components should be weighted differently to optimize learning.
>
> **Comparison across models:**
> |Model|Training Data|RL Method|E-VAds Performance|
> |-|-|-|-|
> |Qwen2.5-VL|General datasets|DPO|Moderate|
> |Keye-VL|Short video domain|GRPO|Moderate|
> |Keye1.5-VL|Short video domain|SAPO|Strong|
>
> Strong performance stems from combining domain-relevant data and advanced sequence-weighted RL training.
>
> > 4. Typos and Cross-Referencing
>
> All typos and cross-referencing inconsistencies will be corrected in the final version.
>
> > 5. Reward Component Selection
>
> MG-GRPO combines three metrics (S, R2, R5) with these principles:
>
> **1. Smooth Guidance for Exploration (R5).** The relaxed metric R5(x) = x provides dense feedback for partially correct traces. For instance, a marginal improvement from x = 0 to x = 0.25 yields a non-zero reward (G(x) ≈ 0.083), preventing the policy from being lost in a "zero-reward landscape" during early stages. This is crucial for open-ended commercial reasoning where binary rewards would provide too sparse supervision.
>
> **2. Non-linear Incentive for Precision (S).** The strict metric S(x) = I(x = 1) creates an uneven reward landscape to penalize "near-misses." While the reward increment from x = 0.5 to x = 0.75 is relatively small, the leap from x = 0.75 (G(x) ≈ 0.417) to a perfect trace (x = 1, G(x) = 1.0) is significantly larger. This non-linear jump compels the model to pursue expert-level precision rather than settling for partially grounded reasoning.
>
> **3. Balanced Intermediate Reward (R2).** R2 bridges the gap between S and R5 by providing moderate rewards for partially correct answers (0.5 for x ∈ {0.75, 0.5}), forcing the model to escape low-quality local optima while maintaining exploration.
>
> By combining these three granularities, G(x) = (S(x) + R2(x) + R5(x)) / 3 provides informative rewards for partially correct traces while strongly favoring fully correct and well-grounded answers. This multi-grained structure enhances group-relative discriminability, enabling the model to capture subtle quality differences among traces within the same group.
>
> [1]. DeepSeekMath: Pushing the Limits of Mathematical Reasoning
> [2]. Delving into RL for Image Generation with CoT: A Study on DPO vs. GRPO
> [3]. ERGO: Efficient High-Resolution Visual Understanding for Vision-Language Models
> [4]. Vision-SR1: Self-Rewarding Vision-Language Model via Reasoning Decomposition and Multi-Reward Policy Optimization

---

> > ### Author Rebuttal · Reviewer_fxMx · 2026-04-02
> >
> > I am satisfied with the author replies to my queries.

---

> > > ### Author Response · Authors · 2026-04-04
> > >
> > > We appreciate your constructive feedback throughout the review process and are delighted that our clarifications have fully resolved your concerns. It is very encouraging to know that our responses satisfactorily addressed your queries and we will incorporate your suggestions to further improve the manuscript.

---

### Official Review · Reviewer_o8SE · 2026-03-12

**Soundness:** 3
**Presentation:** 3
**Significance:** 3
**Originality:** 3
**Overall Recommendation:** 5
**Confidence:** 4

**Summary:**

This paper introduces E-VAds, a benchmark designed for evaluating Multimodal Large Language Models (MLLMs) on e-commerce short video understanding. It first valiadates the uniqueness of e-commerce videos against general videos by computing the information density. Then, it proposes a multi-agent annotation system to collect 19,785 open-ended question-answer pairs, covering various dimensions including perception, cognition, reasoning. It also proposes a reasoning model named E-VAds-R1, which is trained by reinforcement learning with multi-grained rewards. With only a limited number of training samples, this model achieves great performance improvements over baseline methods. Extensive experiments validate the effectiveness of the proposed model.

**Compliance With Llm Reviewing Policy:**

Affirmed.

**Final Justification:**

I keep my rating because I think this paper is very interesting and useful.

**Key Questions For Authors:**

1. During the data collection, how is the video collection and initial filtering pipeline for E-commerce Short Videos conducted? The paper mentions to exclude videos with weak commercial appeal, how to define the weak commercial appeal?

2.. The proposed dataset for reinforcement learning is a little small which contains only 196 videos /980 QA pairs. Is there any method to effectively enlarge the training dataset?

**Limitations:**

1. It looks like that the videos are collected mainly from Taobao platform, which may cause over-fitting and bais. It would be better to show some experiments on cross-platform generalizability of your proposed model.

**Strengths And Weaknesses:**

The proposed data processing pipeline is quite novel because it contains a multi-agent annotation system which can utilize the power of various multimodal large language models. Besides, the proposed multi-grained reward is reasonable. The paper is well-written with sufficient technical details. And the studied topic is quite important because E-commerce short videos present a practical application scenario for multimodal large language models. Considering that existing benchmarks mostly focus on general videos, this work can fill this gap.

---

> ### Author Rebuttal · Authors · 2026-03-31
>
> We thank the reviewer for recognizing the novelty and importance of our work and for acknowledging the value of our "multi-grained reward" design in addressing challenging commercial video understanding tasks. We appreciate your support and address your questions below.
>
> > 1. Video Collection and Initial Filtering Pipeline
>
> The initial filtering uses the following criteria to assess commercial appeal:
> - 7-day click-through rate (CTR)
> - Whether the video is a PPT-style video (excluded)
> - Like-to-view ratio
> - Product category diversity
>
> Videos with low CTR, static PPT formats, or poor engagement metrics are filtered out to ensure high commercial appeal.
>
> > 2. Methods To Enlarge The Training Dataset
>
> We have explored strategies to enlarge the training dataset:
> 1. **Automatic expansion:** Our video collection and filtering pipeline and multi-agent system can generate additional high-quality QA pairs conveniently
> 2. **Data re-partitioning:** We can adjust the train/test split ratio while maintaining benchmark integrity
> 3. **Cross-validation:** We can implement k-fold training to maximize data utilization
>
> We will include discussion of these strategies in the revised manuscript.
>
> > 3. Cross-Platform Generalization
>
> We have conducted transfer learning experiments by evaluating our trained model on VideoAds, an external benchmark for commercial video understanding. The results demonstrate that E-VAds-R1 achieves strong generalization performance:
>
> | Model            | Total Accuracy | Finding Accuracy | Reasoning Accuracy | Summary Accuracy |
> | ---------------- | -------------- | ---------------- | ------------------ | ---------------- |
> | Qwen2.5-VL-7B    | 54.73%         | 62.59%           | 50.69%             | 48.72%           |
> | ReAd-R           | 54.27%         | 60.24%           | 47.66%             | 53.85%           |
> | E-VAds-R1 (Ours) | 55.82%         | 63.29%           | 50.41%             | 51.92%           |
>
> E-VAds-R1 achieves the highest total accuracy (55.82%) among all compared models, with particularly strong performance on Finding Accuracy (63.29%). This validates that the recognition and reasoning capabilities learned from E-VAds transfer effectively to other commercial video understanding tasks. The improvements are especially notable given that VideoAds has a different data distribution and annotation style compared to E-VAds, demonstrating the robustness of our training approach.

---

> > ### Author Rebuttal · Reviewer_o8SE · 2026-04-03
> >
> > Most of my concerns are addressed. I keep my score.

---

> > > ### Author Response · Authors · 2026-04-04
> > >
> > > Thank you for your constructive feedback and your positive evaluation of our work. We are glad our clarifications have addressed your concerns. We will incorporate your suggestions to further improve the manuscript.

---

### Official Review · Reviewer_xjkK · 2026-03-13

**Soundness:** 2
**Presentation:** 3
**Significance:** 4
**Originality:** 3
**Overall Recommendation:** 4
**Confidence:** 4

**Summary:**

This paper tackles the problem of understanding commercial videos. First, the authors distinguish commercial videos from general videos through the lens of information density i.e. commercials have higher visual, audio and text content. Then, the authors propose a benchmark to understand the gaps in existing multimodal-LLMs in understanding and reasoning about commercial videos. Finally, an SFT + RL based fine-tuned protocol is proposed that shows that commercial intent in videos can be understood much better after fine-tuning, but still the performance lags behind the experts' capabilities.

**Compliance With Llm Reviewing Policy:**

Affirmed.

**Final Justification:**

The rebuttal provides sufficient details that helped me upgrade my score:
- ASR-only vs ASR + Visual shows that the benchmark requires vision inputs
- Generalization to other commercial video and importantly non-commercial video benchmarks while initially only in-domain results were presented.
- All the data will be open-sourced

As the authors have acknowledged, the data still suffers from biases despite best mitigation, so I cannot raise my rating even further. I encourage the authors to update the paper with these new findings and discussion on bias.

**Key Questions For Authors:**

Please see the weakness section

**Limitations:**

Limitations section is not provided.

**Strengths And Weaknesses:**

## Strengths:
- This paper is well written and easy to understand.
- The problem space of understanding commercial videos is not much explored in the literature and this paper does a great job of differentiating commercials from general video understanding. The paper quantifies and empirically shows that information density in commercial videos is much higher in Table1. Unsurprisingly, the second and third ranked benchmarks in terms of information density (AdsQA and VideoAds) are also advertisement videos.
- The proposed benchmark has the highest information density compared with previous LLM benchmarks, likely resulting in reasoning challenges to existing models.
- Table2 shows that the gaps between experts and closed-source SOTA LLMs is quite large indicating a nice headroom.
- SFT and RL experiments validate that the benchmark could be improved upon through standard procedures.

## Weaknesses:
- The benchmark and training dataset's QA pairs are automatically generated through a system of multimodal agents, namely Gemini3 Pro and Gemini3 Flash.
  - The quality and difficulty of such QA pairs is unclear. Even-though human experts verify the quality of responses, the inherent bias in question and answer generation may not be overcome by verification alone. Further, using the same LLM system (Gemini) for the whole pipeline can lead to compounding of biases.
  - Example QA pairs are not presented. Supplementary seems to contain only a few videos without QA. It's also unclear how long the questions and answers are.
- No transfer learning from proposed data & model to external benchmarks is shown. This calls into question the generality of the proposal.
- Metric in Table 2 is not clearly introduced. Is it the same as reward in RL? If so, what is the difference between R2 and R3?
- There are many typos. e.g. line 196 and 198 are referring to section 3.1 instead of 4.1. Above usage of R2 vs R3 etc.
- Table4 needs to also show ASR-only to make sure vision inputs are required for the task.
- Overall, while the domain is relatively less explored, all the proposed mechanics such as analyzing information density, orchestration of agents to generate QA, standard sft and rl design are not significant contributions. Together, they do present a coherent theme, however, the above weaknesses are concerning.

---

> ### Author Rebuttal · Authors · 2026-03-31
>
> > 1. Bias in LLM-Generated QA Pairs
>
> We clarify the bias types and mitigation strategies:
>
> **a) Self-Preference and Style Bias:** This refers to LLMs favoring their own generated style, where our multi-agent adversarial system breaks this bias by involving multiple agents with distinct commercial personas (Consumer, Pragmatist, Skeptic, Expert, Creative Director) that ensure question diversity so no single model can dominate.
>
> **b) Modality Bias:** Where dataset design over-relies on a single modality, we enforce strict traceability constraints requiring all reasoning and answers anchored in Visual (V), Audio (A), or OCR (O) evidence, which limits unfounded biases and hallucinations.
>
> **c) Social and Subjective Cognitive Bias:** Stereotypes absorbed during pre-training, where rigorous human review ensures that detection by any of 5 annotators or 2 researchers triggers sample regeneration or discard.
>
> Despite maximum efforts to overcome bias, we acknowledge this remains open, where we will add discussion of this inherent bias to Limitations to alert future researchers.
>
> > 2. Example QA Pairs Not Presented; Length of QA Pairs Unclear
>
> Figure 1(b) and supplementary Figure 14 showcase QA examples across 5 tasks with multi-modal evidence chains, where our generation guidelines enforce strict length limits that require questions to be concise and abstract within 15 Chinese characters and answers to be limited to approximately 30 Chinese characters or 50 English words, as shown in supplementary Figure 10.
>
> > 3. No Transfer Learning to External Benchmarks
>
> We evaluated our trained model on VideoAds, which serves as an external benchmark for commercial video understanding, where transfer learning results demonstrate that E-VAds-R1 achieves strong generalization performance:
> |Model|Total Accuracy|Finding Accuracy|Reasoning Accuracy|Summary Accuracy|
> |-|-|-|-|-|
> |Qwen2.5-VL-7B|54.73%|62.59%|50.69%|48.72%|
> |ReAd-R|54.27%|60.24%|47.66%|53.85%|
> |**E-VAds-R1 (Ours)**|55.82%|63.29%|50.41%|51.92%|
>
> E-VAds-R1 achieves the highest total accuracy (55.82%) and particularly strong Finding Accuracy (63.29%), which validates that recognition and summary capabilities learned from E-VAds transfer effectively to other commercial video tasks. These improvements are especially notable given that VideoAds has a different data distribution and annotation style compared to E-VAds, which demonstrates the robustness of our training approach.
>
> > 4. Typos & Unclear Metric in Table 2 (R2 vs R3)
>
> We apologize for confusion caused by manuscript typos and confirm that "R2" and "R3" denote the same relaxed reward metric defined in Section 5.3, where we will carefully review and correct all identified typos and cross-referencing inconsistencies in the final version.
>
> > 5. Table 4 Needs ASR-Only Ablation
>
> We have conducted the ASR-only ablation study. Results confirm that vision inputs are essential for E-VAds tasks and our E-VAds-R1 has the most significant improvements.
> |Model|Input|S|R3|R5|
> |-|-|-|-|-|
> |Qwen3-VL-8B-Thinking|ASR|0.081|0.328|0.528|
> ||Video|0.160|0.458|0.634|
> ||Video+ASR|0.186|0.482|0.660|
> |Qwen3-Omni-Thinking|ASR|0.092|0.346|0.549|
> ||Video|0.196|0.479|0.659|
> ||Video+ASR|0.207|0.498|0.675|
> |Keye1.5-VL|ASR|0.092|0.338|0.536|
> ||Video|0.179|0.458|0.623|
> ||Video+ASR|0.206|0.482|0.650|
> |E-VAds-R1(Qwen3-VL)|ASR|0.162|0.404|0.597|
> ||Video|0.257|0.532|0.701|
> ||Video+ASR|0.320|0.577|0.736|
>
>
> > 6. Concern About Contribution Significance
>
> We respectfully disagree and summarize our key contributions as follows:
>
> **1. Pioneering Domain Exploration And Commercial Value.** As the reviewer noted, commercial video understanding is "not much explored in the literature." We establish the first dedicated benchmark for e-commerce short videos, filling a critical gap between general video understanding and domain-specific industrial needs.
>
> **2. Multi-modal Information Density Framework.** We identify and quantify the high multi-modal information density characteristic of e-commerce videos through novel metrics (V_den, A_den, O_den), creating a challenging benchmark that pushes beyond current perception-heavy datasets.
>
> **3. Rigorous Data Construction Pipeline.** Our multi-agent annotation system with traceability constraints addresses inherent biases in LLM-generated data, ensuring evidence-grounded QA pairs through adversarial verification and human expert review.
>
> **4. MG-GRPO Training Innovation.** Our multi-grained reward design combines strict and relaxed scoring to provide smooth guidance during early exploration while creating non-linear incentives for expert-level precision—addressing reward in open-ended commercial reasoning.
>
> **5. Strong Empirical Results.** As the reviewer acknowledged, our benchmark creates "reasoning challenges to existing models." E-VAds-R1 achieves substantial improvements over strong open-source baselines, establishing a new frontier for the community to build upon.

---

> > ### Author Rebuttal · Reviewer_xjkK · 2026-04-04
> >
> > ```Figure 1(b) and supplementary Figure 14 showcase QA examples across 5 tasks with multi-modal evidence chains, where our generation guidelines enforce strict length limits that require questions to be concise and abstract within 15 Chinese characters and answers to be limited to approximately 30 Chinese characters or 50 English words, as shown in supplementary Figure 10.```
> >
> > Will the data be open-sourced?
> >
> > ```We evaluated our trained model on VideoAds, which serves as an external benchmark ```
> >
> > How does the trained model perform on non-commercial video understanding benchmarks like LVBench, Video-MME etc. compared with baseline?
> >
> > Thanks for the ASR ablation. It is appreciated and looks good.

---

> > > ### Author Response · Authors · 2026-04-05
> > >
> > > Thank you for your thoughtful follow-up questions. We appreciate the opportunity to clarify these important points.
> > >
> > > > **Q1: Will the data be open-sourced?**
> > >
> > > **Yes**, we commit to open-sourcing the **E-VAds dataset, the multimodal information density analysis framework and the trained model E-VAds-R1 upon paper acceptance.** Specifically, **all 3,961 videos** and **19,785 QA pairs** will be publicly released, and **we have already obtained appropriate authorization from Taobao** to use these e-commerce videos for academic research, which ensures that the dataset will be distributed under a license that fully respects the platform terms of service.
> > >
> > >
> > > > **Q2: Performance on non-commercial video benchmarks.**
> > >
> > > We evaluated E-VAds-R1 on Video-MME to assess generalization to general-purpose video understanding, and the results show **consistent improvements over the base model** Qwen2.5-VL 7B:
> > >
> > > | Video-MME | E-VAds-R1 | Qwen2.5-VL 7B | Δ          |
> > > | --------- | --------- | ------------- | ---------- |
> > > | Short     | 73.33%    | 70.44%        | **+2.89%** |
> > > | Medium    | 59.78%    | 58.56%        | **+1.22%** |
> > > | Long      | 49.56%    | 46.56%        | **+3.00%** |
> > > | Overall   | 60.89%    | 58.52%        | **+2.37%** |
> > >
> > > These improvements can be attributed to the uniquely high multimodal information density that characterizes our E-VAds dataset. Because e-commerce short videos contain substantially denser visual, audio and textual signals than general-purpose or even other commercial video datasets, models trained on E-VAds learn to extract and integrate fine-grained cues under high-information-load conditions, which in turn strengthens their fundamental perception and reasoning capabilities that transfer to broader video understanding tasks.
> > >
> > > The fact that E-VAds-R1 outperforms Qwen2.5-VL 7B across all video length categories on Video-MME demonstrates that our training protocol does not compromise generalization to non-commercial domains, and it further indicates that the multi-grained reward design in MG-GRPO encourages the model to develop more robust temporal reasoning and evidence grounding skills that benefit both domain-specific and general video understanding.
> > >
> > > We hope these additional results adequately address your concerns. **We would be deeply grateful if you would consider raising your score in light of this new evidence.**

---

### Decision · Program_Chairs · 2026-04-30

**Decision:**

Accept (regular)

**Comment:**

This paper introduces E-VAds, a benchmark for e-commerce short video understanding, along with an RL-based model to address commercial intent reasoning. Reviewers found the problem setting meaningful and under-explored, and appreciated the dataset construction, density analysis, and strong empirical results.

A key concern raised by one reviewer was the limited domain scope and the need for broader generalization to other datasets. This issue was also discussed by other reviewers, and the authors provided additional experiments on external benchmarks as well as further analysis clarifying the role of RL versus SFT. These responses largely addressed the generalization concern, and were acknowledged by the reviewers.

Overall, the paper presents a valuable benchmark and a coherent study of a challenging domain, with most concerns adequately addressed during rebuttal. I recommend accept.